

**Carbon cycle feedbacks in an idealized and a scenario simulation of negative emissions in CMIP6**
**Earth system models**
Ali Asaadi[1], Jörg Schwinger[1], Hanna Lee[1,2], Jerry Tjiputra[1], Vivek Arora[3], Roland Séférian[4], Spencer
Liddicoat[5], Tomohiro Hajima[6], Yeray Santana-Falcón[4] , Chris D. Jones[5]
[1]NORCE Norwegian Research Centre & Bjerknes Centre for Climate Research, Bergen, Norway
[2]Department of Biology, Norwegian University of Science and Technology, Trondheim, Norway
[3]Canadian Centre for Climate Modelling and Analysis, Environment and Climate Change Canada,
Victoria, BC, Canada
[4]CNRM, Université de Toulouse, Meteo-France, CNRS, Toulouse, France
[5]Met Office Hadley Centre, Exeter, United Kingdom
[6]Research Institute for Global Change, Japan Agency for Marine-Earth Science and Technology,
Yokohama 236-0001, Japan
[*]Corresponding author, ali.asaadi@mail.mcgilll.ca
**Abstract**
Limiting global warming to 1.5°C by the end of the century is an ambitious target that requires
immediate and unprecedented emission reductions. In the absence of sufficient near term mitigation,
this target will only be achieved by carbon dioxide removal (CDR) from the atmosphere later during this
century, which would entail a period of temperature overshoot. Next to the socio-economic feasibility
of large-scale CDR, which remains unclear, the effect on biogeochemical cycles and climate are key to
assessing CDR as a mitigation option. Changes in atmospheric $CO_2$ concentration and climate alter the
$CO_2$ exchange between the atmosphere and the underlying carbon reservoirs of land and the ocean.
Here, we investigate carbon cycle feedbacks under idealized and more realistic overshoot scenarios in
an ensemble of Earth system models. The response of oceanic and terrestrial carbon stocks to changes
in atmospheric $CO_2$ concentration and changes in surface climate (the carbon-concentration and
carbon-climate feedback, quantified by the feedback metrics $\beta$ and $\gamma$, respectively) show a large
hysteresis. This hysteresis leads to growing absolute values of $\beta$ and $\gamma$ during phases of negative
emissions. We find that this growth is spatially quite homogeneous, since the spatial patterns of
feedbacks do not change significantly for individual models. We confirm that the $\beta$ and $\gamma$ feedback
metrics are a relatively robust tool to characterize inter-model differences in feedback strength since
the relative feedback strength remains largely stable between phases of positive and negative
emissions and between different simulations, although exceptions exist. When emissions become
negative, we find that the model uncertainty (model disagreement) in $\beta$ and $\gamma$ increases stronger than
expected from the assumption that the uncertainties would accumulate linearly with time. This
indicates that the model response to a change from increasing to decreasing forcing introduces an
additional layer of uncertainty, at least in idealized simulations with a strong signal. We also briefly
discuss the existing alternative definition of feedback metrics based on instantaneous carbon fluxes
instead of carbon stocks and provide recommendations for the way forward and future model
intercomparison projects.



## 1. Introduction

Estimated remaining carbon budgets compatible with limiting anthropogenic warming to 1.5 or 2 °C above pre-industrial levels are extremely tight and will be exhausted within the next few decades if the current emission rate is maintained (e.g., Rogelj et al. 2015; Goodwin et al. 2018; V. Masson-Delmotte et al. 2018). Therefore, unless $CO_2$ emissions are reduced immediately at an unprecedented rate, the 1.5 or 2°C targets can only be reached after a period of temperature overshoot (Rogelj et al. 2015; Ricke et al. 2017; Geden and Löschel 2017; Riahi et al. 2021). Although the option to remove large quantities of carbon from the atmosphere remains speculative (Gasser et al. 2015; Smith et al. 2016; Larkin et al. 2018; Fuss et al. 2018; Creutzig et al. 2019), in such overshoot pathways, too large near-term carbon emissions would be compensated by large-scale carbon dioxide removal (CDR) later in this century. Research on negative emissions exploring the reversibility of $CO_2$-induced climate change has been conducted for more than a decade (e.g., Boucher et al. 2012; Wu et al. 2015; Tokarska and Zickfeld 2015; Li et al. 2020; Jeltsch-Thömmes et al. 2020; Yang et al. 2021; Schwinger et al. 2022; Bertini and Tjiputra 2022). These studies generally report a hysteresis behavior of the Earth system under negative emission, resulting in greatly varying reversibility for different aspects of the Earth system. While the surface temperature trend follows a reduction in atmospheric $CO_2$ relatively closely (e.g., Boucher et al. 2012; Jeltsch-Thömmes et al. 2020), hysteresis can be large in the interior ocean, making for example ocean heat content and steric sea level rise as well as interior ocean oxygen content and acidification largely irreversible on policy relevant timescales (Mathesius et al. 2015; Li et al. 2020; Schwinger et al. 2022; Bertini and Tjiputra 2022). The same is true for the loss of carbon from thawing permafrost soils (MacDougall et al. 2015; Gasser et al. 2018; Park and Kug 2022; Schwinger et al. 2022).

Carbon emissions drive multiple responses of the Earth system via changes in its physical climate and the biogeochemical carbon cycle. Under increasing atmospheric $CO_2$ concentrations, increasing carbon uptake by the ocean and terrestrial biosphere slows down global climate change by removing the greenhouse gas $CO_2$ from the atmosphere, a process that is mainly driven by the dissolution of $CO_2$ into the oceans (e.g. Revelle and Suess 1957, Siegenthaler and Oeschger 1978) and the $CO_2$-fertilisation effect on the terrestrial biosphere (Schimel et al. 2015). On the other hand, Earth system model (ESM) simulations show that this carbon uptake is reduced by progressive global warming due to, among others, changes in ocean circulation and a reduction of $CO_2$ solubility in warmer waters, as well as increased respiration rates from soils (Tharammal et al. 2019; Arora et al. 2020; Canadell et al. 2021), and carbon release from degrading permafrost. These two feedback processes, the response to rising $CO_2$ concentrations and the response to climate change, are termed carbon-concentration and carbon-climate feedback, respectively (Gregory et al. 2009). In the context of overshoot pathways, carbon cycle feedbacks determine the efficiency of negative emissions as the oceans and the terrestrial biosphere will first take up carbon at reduced rates and eventually turn into sources of carbon to the atmosphere (Jones et al. 2016a; Schwinger and Tjiputra 2018).

The carbon-concentration and carbon-climate feedbacks can be characterized by feedback metrics, for example, by feedback factors $\beta$ and $\gamma$ (Friedlingstein et al. 2003) that quantify the gain/loss of carbon in terrestrial or marine reservoirs per unit change in atmospheric $CO_2$ concentration and temperature, respectively (see Section 2 for details). These feedback factors are valuable tools to compare the feedback strength among different models (Friedlingstein et al. 2003, 2006; Yoshikawa et al. 2008; Boer



and Arora 2009; Gregory et al. 2009; Roy et al. 2011; Arora et al. 2013, 2020) and can be calculated
using idealized model simulations, in which the effect of $CO_2$ on the carbon cycle and the radiative effect
of $CO_2$ are decoupled. In a biogeochemically coupled (BGC) simulation, the radiation code of an ESM
does not respond to increasing atmospheric $CO_2$ concentrations, but the terrestrial and marine carbon
cycles do. There is little climate change in such a simulation, which can therefore be used to quantify
the carbon-concentration feedback. The difference between a standard (fully coupled, COU) simulation
and the BGC simulation is used to quantify the carbon-climate feedback. In the last two phases of the
Coupled Model Intercomparison Project (CMIP5 and CMIP6, Taylor et al. 2012; Eyring et al. 2016)
carbon cycle feedbacks were addressed by conducting additional decoupled simulations of the standard
1% $CO_2$ simulation (1pctCO2 hereafter), which prescribes an increase in atmospheric $CO_2$ by 1% per
year until quadrupling (Arora et al. 2013, 2020). Next to this idealized simulation, the protocol for the
CMIP6 Coupled Climate-Carbon Cycle Model Intercomparison Project (C4MIP, Jones et al. 2016b) also
proposes a BGC simulation for the SSP5-3.4-OS scenario (O'Neill et al. 2016). This scenario describes an
overshoot pathway, in which emissions increase unmitigated until 2040, but strong mitigation
(including CDR) is undertaken thereafter. In contrast to the 1pctCO2 simulation, where no forcing other
than atmospheric $CO_2$ is varied, the quantification of feedbacks in this scenario simulation is
complicated by the presence of land use change and changes in radiative forcing through emissions of
aerosols and non-$CO_2$ greenhouse gasses (Melnikova et al. 2021, 2022).
Permafrost soils in the northern high latitudes have accumulated roughly 1100-1700 Pg of carbon in
the form of frozen organic matter, about twice as much as currently contained in the atmosphere
(Hugelius et al. 2014; Schuur et al. 2015). The release of $CO_2$ and methane ($CH_4$) from thawing
permafrost will amplify global warming due to anthropogenic emissions and further accelerate
permafrost degradation and microbial decomposition (Feng et al. 2020; Smith et al. 2022). This positive
feedback and the fact that Arctic temperatures are increasing twice as fast as the global average
(Jenkins and Dai 2021; Liang et al. 2022) have made permafrost to be considered among the key tipping
elements of the climate system, although it may not be an abrupt but irreversible process (Armstrong
McKay et al. 2022; Yokohata et al. 2020; Lenton et al. 2019). A temporary temperature overshoot might
entail important legacy effects as high latitude ecosystems and the state of permafrost-affected soils
take several centuries to adjust to the new atmospheric condition (de Vrese and Brovkin 2021). Current
generation ESMs are still in their infancy when it comes to representing the complex and often small-
scale processes of permafrost carbon degradation. Here we take advantage of the fact that one of the
CMIP6 ESMs considered in this study has a vertically resolved representation of soil carbon, which
enables us to estimate the contribution of permafrost degradation to the total carbon-climate feedback
for this model.
Except for the recent studies by Schwinger and Tjiputra (2018) and Melnikova et al. (2021, 2022) all
previous studies that quantify carbon-concentration and carbon-climate feedbacks are based on ESM
simulations with increasing atmospheric $CO_2$. Here, we take advantage of a simulation conducted for
the CMIP6 Carbon Dioxide Removal Model Intercomparison Project (CDRMIP, Keller et al. 2018) that
mirrors the 1pctCO2 simulation by prescribing a decrease of atmospheric $CO_2$ by 1% per year (1pctCO2-
cdr). We complement this simulation with a BGC simulation (1pctCO2-cdr-bgc) to quantify, in a manner
consistent with previous feedback studies (Arora et al. 2013, 2020), carbon-concentration and carbon-



climate feedbacks under negative emissions in an ensemble of CMIP6 ESMs. We complement these previous studies by a spatial analysis of feedback patterns, and compare the feedbacks from the positive and negative emission phases of the 1pctCO2 and 1pctCO2-cdr simulations to the feedbacks obtained from the SSP5-3.4-OS scenario. For the latter, land use change has been shown to have a dominant effect over carbon-concentration or carbon-climate feedbacks by Melnikova et al. (2021, 2022), and these authors present a more detailed analysis of the role of land use change in the SSP5-3.4-OS scenario. Since land use change is not a feedback process, we focus in this study on regions that are not dominated by agricultural areas when comparing feedbacks between the SSP5-3.4-OS and 1pctCO2 simulations.

The purpose of this study is to investigate the evolution of carbon cycle feedbacks and their uncertainty under deployment of negative emissions. Since feedback metrics are known to depend on the emission (or concentration) pathway, we investigate the relative feedback strength and the spatial patterns of feedbacks between positive and negative emission phases as well as between idealized and scenario simulations. We also briefly explore the contribution of permafrost carbon losses to the carbon-climate feedback and the impact of alternative feedback metric definitions in the context of negative emissions.

## 2. Description of feedback metrics, simulations, and models

### 2.1 Carbon cycle feedback metrics

The sensitivity of the carbon cycle to changes in atmospheric $CO_2$ concentration ($[CO_2]$) and its sensitivity to changes in physical climate can be measured using two feedback metrics, which can be based on either changes in carbon stocks (as introduced by Friedlingstein et al., 2003) or instantaneous carbon fluxes (as introduced by Boer and Arora 2009). Changes in carbon stocks are equivalent to the time-integrated carbon fluxes across the air-land and air-sea interfaces, such that for the Friedlingstein et al. approach (referred to as integrated flux-based approach), the two feedback metrics are:

1. $\beta$ (PgC/ppm), which quantifies the strength of the carbon-concentration feedback, i.e., the changes in oceanic and terrestrial carbon stocks ($\Delta C_{L,O}$) in response to changes in atmospheric $CO_2$ concentration ($\Delta[CO_2]$), and
2. $\gamma$ (PgC/°C), which measures the strength of the carbon-climate feedback, i.e., changes in land and ocean carbon stocks ($\Delta C_{L,O}$) in response to changes in global average surface temperature ($\Delta T$), where $\Delta T$ serves as a proxy for climate change.

Carbon feedback analysis requires, in addition to a standard fully coupled (COU) simulation, a biogeochemically (BGC) coupled simulation. In a BGC simulation, atmospheric $[CO_2]$ is kept constant at its pre-industrial values for the radiative transfer calculations, to isolate the response of land and ocean biogeochemistry to rising $[CO_2]$ in the absence of $CO_2$-induced climate change. Using this pair of simulations (COU and BGC) results in the following expressions for $\beta$ and $\gamma$ (see Schwinger et al. 2014 for a derivation).



$$\beta_X = \frac{1}{\Delta[CO_2]}\left(\frac{\Delta C_X^{BGC}\Delta T^{COU} - \Delta C_X^{COU}\Delta T^{BGC}}{\Delta T^{COU} - \Delta T^{BGC}}\right)$$
$$\simeq \frac{\Delta C_X^{BGC}}{\Delta[CO_2]} \tag{1}$$

$$\gamma_X = \frac{\Delta C_X^{COU} - \Delta C_X^{BGC}}{\Delta T^{COU} - \Delta T^{BGC}}$$
$$\simeq \frac{\Delta C_X^{COU} - \Delta C_X^{BGC}}{\Delta T^{COU}} \tag{2}$$

where $X$ can be either $L$ for land or $O$ for ocean. Although there is no change in the radiative forcing of
$CO_2$ in the BGC simulation (such that we could expect $\Delta T^{BGC} = 0$), surface temperature can vary due
to changes in other radiative forcing agents (aerosols and non-$CO_2$ greenhouse gasses). Even in the
idealized 1pctCO2 simulation, where $CO_2$ is the only variable forcing, there are some climatic changes
over the land surface due to a reduction in latent heat fluxes associated with stomatal closure at higher
$CO_2$ levels, as well as changes in vegetation structure, coverage, and composition (Arora et al. 2020),
which result in a small temperature increase along with changes in precipitation and soil moisture. The
assumption of $\Delta T^{BGC} = 0$ will simplify equations (1) and (2) such that the rightmost term holds. The
instantaneous flux-based approach is equivalent to equations (1) to (2) except that the deviation of the
carbon pools $\Delta C_X$ are replaced by the instantaneous air-sea or air-land carbon fluxes $F_X$. To distinguish
these feedback metrics from the integrated flux-based ones, the capital letters B and Γ are used to
denote them. The units of B and Γ are PgCyr$^{-1}$ppm$^{-1}$ and PgCyr$^{-1}$°C$^{-1}$, respectively.
It is worth mentioning that these idealized feedback frameworks should be seen as a technique for
assessing the relative sensitivities of models and understanding their differences (i.e. the model
uncertainty of the estimated feedbacks), rather than as absolute measures of invariant system
properties (Gregory et al. 2009; Ciais et al. 2013). Therefore, the values of carbon cycle feedback metrics
can vary over time within a model simulation (e.g. Arora et al. 2013) or between different scenarios
(Hajima et al. 2014).
To gain insight into the reasons for differing responses among models, we apply the decomposition of
the simplified expression for $\beta_L$ (Eq. 1, assuming $\Delta T^{BGC} = 0$) following Arora et al. (2020). This allows
us to investigate the contributions from different processes to changes in vegetation and soil carbon
reservoirs ($\Delta C_V$ and $\Delta C_S$, respectively).
$$\beta_L = \frac{\Delta C_L^{BGC}}{[CO_2]} = \frac{\Delta C_V^{BGC} + \Delta C_S^{BGC}}{[CO_2]} = \left(\frac{\Delta C_V^{BGC}}{\Delta NPP^{BGC}}\frac{\Delta NPP^{BGC}}{\Delta GPP^{BGC}}\frac{\Delta GPP^{BGC}}{[CO_2]}\right) + \left(\frac{\Delta C_S^{BGC}}{\Delta R_h^{BGC}}\frac{\Delta R_h^{BGC}}{\Delta LF^{BGC}}\frac{\Delta LF^{BGC}}{[CO_2]}\right)$$
$$= \tau_{cveg\Delta}CUE_\Delta\frac{\Delta GPP^{BGC}}{[CO_2]} + \tau_{csoil\Delta}\frac{\Delta R_h^{BGC}}{\Delta LF^{BGC}}\frac{\Delta LF^{BGC}}{[CO_2]} \tag{3}$$

ΔNPP, ΔGPP, Δ$R_h$, and ΔLF represent deviations of the net primary productivity, gross primary
productivity, heterotrophic respiration, and litterfall flux, respectively, from their pre-industrial values.
The terms $\tau_{cveg\Delta}$ and $\tau_{csoil\Delta}$ are turnover times (in years) of carbon in the vegetation and litter plus



soil pools. $\frac{\Delta NPP}{\Delta GPP}$ is a measure of carbon use efficiency for the fraction of GPP (above its pre-industrial
value) that turned into NPP after subtracting autotrophic respiration losses (denoted as $CUE_\Delta$).
$\frac{\Delta GPP}{[CO_2]}$(PgCyr$^{-1}$ppm$^{-1}$) and $\frac{\Delta R_h}{\Delta LF}$ are a measure of the global $CO_2$ fertilization effect and the increase in
heterotrophic respiration per unit increase in litterfall rate, respectively. Also, $\frac{\Delta LF}{[CO_2]}$ (PgCyr$^{-1}$ppm$^{-1}$)
measures the global increase in litterfall rate per unit increase in $CO_2$.
**2.2 Model simulations**
The 1pctCO2 experiment is a highly idealized model experiment that prescribes a rate of 1% per year
increase in [$CO_2$] from pre-industrial values until quadrupling after 140 years. No other forcings are
varied in this experiment, i.e., land use as well as non-$CO_2$ greenhouse gasses and aerosol
concentrations are held constant at their pre-industrial levels. This experiment has already been
performed by the first coupled atmosphere-ocean general circulation models in the late 1980s, and
important climate metrics such as the transient climate response (TCR; Meehl et al. 2020) and the
transient response to cumulative emissions (TCRE; e.g. Gillett et al. 2013) are formally defined through
the 1pctCO2 simulation. Likewise, the C4MIP carbon cycle feedback analysis for the last two phases of
CMIP (Arora et al. 2013, 2020) relied on this simulation. Given the importance of the 1pctCO2
experiment, the CMIP6 CDRMIP protocol proposes an experiment that mirrors the 1pctCO2 simulation
by starting from its endpoint at year 140 and decreasing atmospheric $CO_2$ at a rate of 1% per year until
pre-industrial [$CO_2$] is restored (1pctCO2-cdr). This experiment is designed to investigate the response
of the Earth system to carbon dioxide removal in an idealized fashion. We note that the implied rates
of CDR in the 1pctCO2-cdr simulation are huge and not practically feasible. Also, there is a jump from
very large positive to very large negative diagnosed emissions at the end of year 140, which is clearly
unrealistic. To investigate carbon cycle feedbacks under CDR, we have complemented the 1pctCO2-cdr
simulation with a biogeochemical coupled 1pctCO2-cdr-bgc simulation that starts from the endpoint of
the 1pctCO2-bgc simulation at year 140.
The family of Shared Socioeconomic Pathways (SSPs, O'Neill et al. 2014) is designed to represent
different socio-economic futures that social, demographic, political, and economic developments could
lead to. These narrative SSPs are the basis for a set of quantitative future scenarios, a selection of which
is now being used as input for scenario simulations by the latest ESMs in the frame of the CMIP6
ScenarioMIP (O'Neill et al. 2016). The SSP5-3.4-OS scenario follows the high emission SSP5-8.5 pathway
until 2040 at which point strong mitigation policies are introduced to rapidly reduce emissions to zero
by about 2070 and to net-negative levels thereafter (Fig. 3 of O'Neill et al. 2016). In contrast to the
1pctCO2 simulation, the SSP5-3.4-OS scenario includes land use change as well as time varying forcing
from aerosols and non-$CO_2$ greenhouse gasses throughout the simulation period (Fig. 1 of Liddicoat et
al. 2021). For this study, we use the 1pctCO2, 1pctCO2-cdr, and SSP5-3.4-OS simulations from the
CMIP6 archive together with the corresponding biogeochemically coupled simulations of these
experiments. We note that the biogeochemically coupled 1pctCO2-cdr-bgc experiment is not part of
CMIP6, but has been performed for this study by participating modelling groups.



The C4MIP simulation protocol does not allow to separate carbon release (or uptake) through land use
changes from the carbon-concentration feedback, since land use is active in the biogeochemically
coupled SSP5-3.4-OS simulation. To focus on carbon cycle feedbacks, we eliminate the effect of land
use changes as much as possible by identifying regions that are mostly unaffected by human activity
(referred to as "natural land"). To accomplish this in a way that available CMIP6 output permits, we
define natural land as grid cells with a maximum (over the period 2015 to 2100) crop-land fraction of
less than 25%. The threshold of 25% used here for our heuristic approach, is a compromise between
accuracy (some signal of land use change is still present) and spatial coverage (with increasingly lower
thresholds, larger areas of the globe are excluded). Our results are not very sensitive to variations in
the threshold between approximately 10 and 30%. Maps of maximum SSP5-3.4-OS cropland fraction
for each model (Fig. S1) indicates that a 25% threshold reasonably identifies hotspots of agricultural
production. To make our analysis comparable between the SSP5-3.4-OS and 1pctCO2 simulations, we
use the same set of grid cells also for the 1pctCO2 simulation (unless otherwise noted), even though
land cover is not changed from its pre-industrial state in this simulation.
**2.3 Participating Earth System Models**
Table 1 summarizes the five ESMs that contributed to this study along with the experiments used for
the analyses presented here. The primary features of these models are listed in Table 2 of Arora et al.
(2020). MIROC-ES2L, NorESM2-LM (which employs version 5 of the Community Land Model, CLM5),
and UKESM1-0-LL have a representation of the terrestrial nitrogen cycle implemented and coupled to
their carbon cycle. Only the UKESM1-0-LL model has a land component that dynamically simulates
vegetation cover and competition between their plant functional types (PFTs). Fire is included in the
CNRM-ESM2-1 and NorESM2-LM models. NorESM2-LM, is the only ESM with vertically resolved soil
carbon, which allows studying the impact of warming on the carbon stored in permafrost soils in more
detail. In this study, the gridcell was considered permafrost where the pre-industrial maximum active
layer thickness was shallower than three meters.

**Table 1:** List of CMIP6 ESMs used in this study, names of their biogeochemical component models, resolution
and experiment variants used.

|  | **CanESM5** | **CNRM-ESM2-1** | **MIROC-ES2L** | **NorESM2-LM** | **UKESM1-0-LL** |
|---|---|---|---|---|---|
| **Atmosphere and land resolution** | 2.81°x2.81°[*] | 1.4°x1.4° | 2.81°x2.81° | 1.9°x2.5° | 1.875°x1.25° |
| **variant** | r1i1p1f1 & r1i1p2f1[*] | r1i1p1f2 | r1i1p1f2 | r1i1p1f1 | r4i1p1f2 & r1i1p1f2 |
| **Ocean resolution** | 1° (finer in the tropics) | 1° (finer in the tropics) | 1° (finer close to North Pole and Equator) | 1° (finer near the Equator) | 1° |
| **Ocean model name** | CMOC (biology); carbonate | PISCESv2-gas | OECO2 | iHAMOCC | MEDUSA-2.1 |



| | | | | | |
|---|---|---|---|---|---|
| | chemistry follows OMIP protocol | | | | |
| **Land model name** | CLASS-CTEM | ISBA–CTRIP | MATSIRO (physics), VISIT-e (BGC) | CLM5 | JULES-ES-1.0 |
| **Reference** | Swart et al. (2019) | Séférian et al. (2019) | Hajima et al. (2020) | Tjiputra et al. (2020); Seland et al. (2020) | Sellar et al. (2019) |

*CMIP6 experiment variant used across different simulations including: piControl, historical, hist-bgc, ssp585, ssp585-bgc,
ssp534-over, ssp534-over-bgc, 1pctCO2, 1pctCO2-bgc, 1pctCO2-cdr, and 1pctCO2-cdr-bgc experiments.

**3. Results and Discussion**
**3.1 Atmospheric $CO_2$, temperature, and carbon fluxes**
The atmospheric $CO_2$ concentration ($[CO_2]$) for the concentration-driven SSP5-3.4-OS scenario, peaks
at 571 ppm (a doubling of pre-industrial $CO_2$ concentration) in the year 2062 and decreases to  497 ppm
in 2100 (Fig. 1a). According to the scenario design (see O'Neill et al. 2016), strong mitigation policies
(including deployment of bioenergy with carbon capture and storage (BECCS) and other carbon dioxide
removal technologies) start in 2040 resulting in an immediate decrease in the $CO_2$ growth rate that
peaks in 2041 (Fig. 1e). In the 1pctCO2 simulation, the prescribed $[CO_2]$ is symmetric around its $4xCO_2$
peak of 1133 ppm in the year 140 (Fig. 1c). The rate of change of the $CO_2$ concentration (Fig. 1e) is very
different between SSP5-3.4-OS and 1pctCO2 experiments. In particular, the $CO_2$ growth rate in the
idealized 1pctCO2 experiment has a sudden and large jump from positive to negative values at the
transition from the ramp-up to the ramp-down phase.
The five participating ESMs show large differences in global mean surface air temperature change,
relative to pre-industrial values, under the SSP5-3.4-OS simulation (Fig. 1b). Peak temperatures vary
from 2°C in NorESM2-LM to 4.35°C in CanESM5. The timing of the global surface air temperature peak
varies from 2062 for the MIROC-ES2L and UKESM1-0-LL models to 2100 for CNRM-ESM2-1. After this
peak, the temperature declines again (except for CNRM-ESM2-1) reaching end-of-the-century values
that range from 1.39°C above pre-industrial in NorESM2-LM to 3.47°C in CanESM5. The multi-model
mean global surface air temperature is 2.66°C at the end of the 21st century. The model-mean growth
rate of the global surface air temperature (Fig. 1f) plateaues at about 0.05°C/yr between approximately
2030-2050 before it starts to decline to below zero towards the end of the simulation.
Temperature changes in the BGC simulation of SSP5-3.4-OS are not negligible since the non-$CO_2$ forcing
agents as well as land use change do evolve in time in this scenario, in contrast to the idealized 1pctCO2
simulation. Positive peak temperature anomalies range from 0.37°C (CNRM-ESM2-1 in 2098) to 1.29°C
(CanESM5 in 2057). UKESM1-0-LL also shows a pronounced negative temperature anomaly during the
historical period of the BGC simulation of -0.80°C in the year 1990.

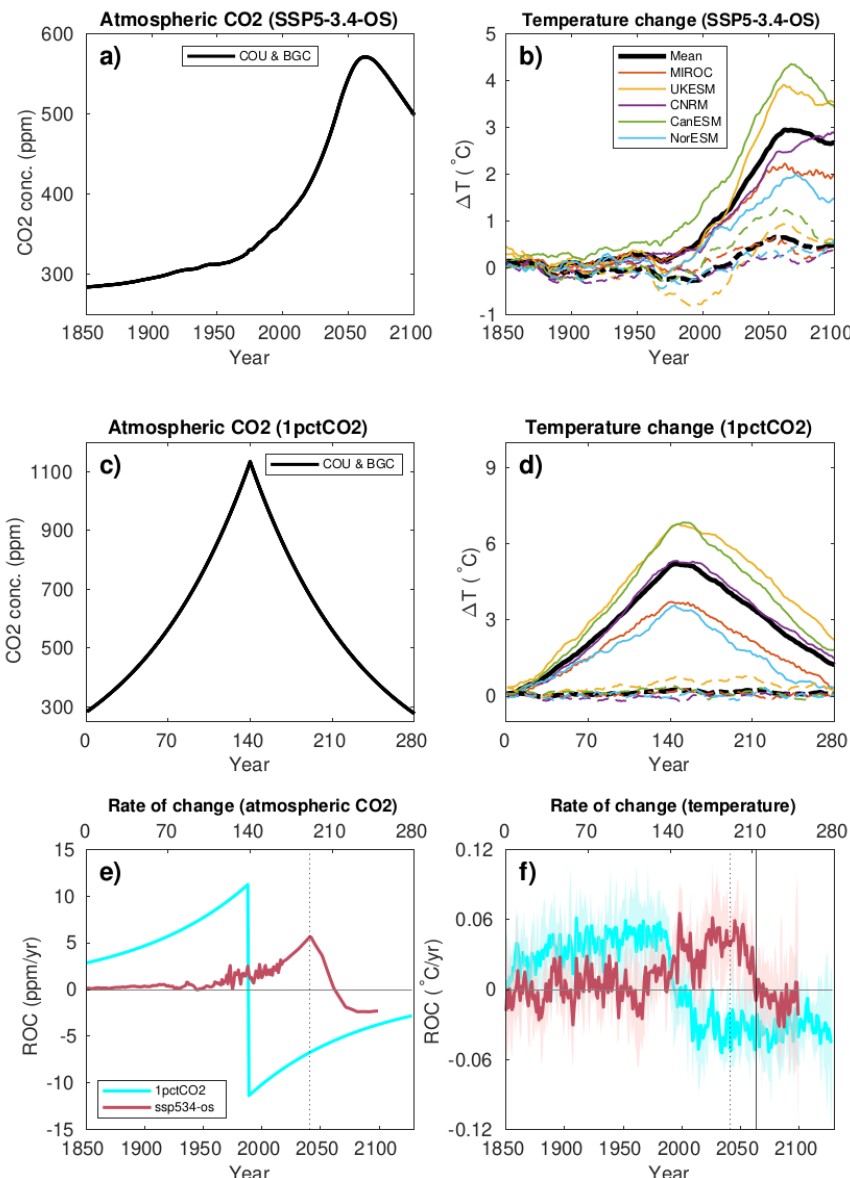


**Figure 1:** Atmospheric $CO_2$ concentration and surface air temperature changes in the fully coupled (solid
lines) and biogeochemically coupled (dashed lines) configurations of the SSP5-3.4-OS (a,b) and 1pctCO2
(c,d) experiments. The rates of change in the prescribed atmospheric $CO_2$ concentrations is shown in
panel e, and the model mean rate of surface temperature change from the fully coupled simulations is
shown in panel f. The dotted (solid) vertical lines in panels e and f indicate the peak of the $CO_2$ growth
rate ($CO_2$ concentration) in the SSP5-3.4-OS scenario. Shadings in panel f show the range across the
models. An 11-year moving average has been used in panels b, d, and f.




In the 1pctCO2 simulation, the peak temperature anomalies vary from 3.57°C (in year 144) in NorESM2-
LM to 6.84°C (in year 151) in CanESM5 (Fig. 1d). Thereafter, temperature anomalies decline to values
ranging from 0.29°C in NorESM2-LM to 2.2°C in UKESM1-0-LL at the end of the ramp-down period (year
280). The 1pctCO2 BGC simulation shows, compared to the SSP5-3.4-OS BGC simulation, smaller
temperature anomalies ranging from -0.22°C (CNRM-ESM2-1 in year 149) to 0.79°C (UKESM1-0-LL in
year 207). The smaller magnitude of the temperature anomaly in the BGC simulation of the SSP5-3.4-
OS scenario compared to the 1pctCO2-BGC simulations (also given the much higher $CO_2$ forcing in the
latter) suggests that a substantial part of the carbon-climate feedback in the SSP5-3.4-OS scenario might
be caused by non-$CO_2$ forcings.
For atmosphere-land fluxes, our analysis is complicated by the fact that land use changes are present
in the SSP5-3.4-OS scenario. Here, we focus on comparing fluxes and feedbacks for grid cells that are
dominated by "natural land" (see Sec. 2.2 for more details). Note that, for comparability, we consider
the same set of grid cells in the 1pctCO2 simulation, even though land cover stays at its pre-industrial
state in this simulation. In the SSP5-3.4-OS simulations, the model-mean annual $CO_2$ fluxes (Fig. 2)
continue rising until the rate of change of [$CO_2$] reaches its peak in 2041. After the peak, natural
atmosphere-land and atmosphere-ocean fluxes start to decline rapidly in all models with little time lag.
UKESM1-0-LL and MIROC-ES2L simulate negative fluxes (i.e., natural land turns into a carbon source)
before the end of the 21st century in the COU simulation (Fig. 2a). Without the effect of $CO_2$ induced
warming (BGC simulation, Fig. 2b), only MIROC-ES2L shows a significant carbon source from the
terrestrial biosphere before 2100, while the model-mean still shows a sink. In the fully coupled 1pctCO2
experiment, sink-to-source transition of the terrestrial biosphere occurs around year 165 in the model
mean, 25 years after the rate of change of [$CO_2$] peaks (Fig. 2c). Consistent with what is seen in the
biogeochemically coupled SSP5-3.4-OS, the sink-to-source transition occurs 10 years later without the
effect of warming in the 1pctCO2-BGC experiment. However, the terrestrial $CO_2$ source at the end of
the biogeochemically coupled 1pctCO2 simulation is *larger* than in the fully coupled simulation. We also
observe (Fig. 2c,d) that models which take up more (less) terrestrial carbon during the $CO_2$ ramp-up
phase (1pctCO2) release more (less) carbon towards the end of the $CO_2$ ramp-down phase (1pctCO2-
cdr-bgc). We therefore interpret the increased source of carbon at the end of the 1pctCO2-BGC
simulation as an outgassing of the excess carbon that could be taken up in the absence of climate
warming. The net negative emission phase of the SSP5-3.4-OS scenario is too short to show this effect
in 2100 (where the warming effect still *increases* the terrestrial carbon source).
Likewise, the warming of the world's oceans in both simulations, tends to reduce the carbon uptake or
increase the oceanic carbon source. The model spread for atmosphere-ocean carbon fluxes (Fig. 2,
panels e to h) appears to be much smaller than for the atmosphere-land fluxes. In the SSP5-3.4-OS
simulation, the ocean remains a sink of carbon in all models until the end of the simulation in 2100. In
the 1pctCO2 simulation the ocean turns into a source of $CO_2$ to the atmosphere around year 175, and
in the BGC simulation without warming this transition is delayed by 7 years.



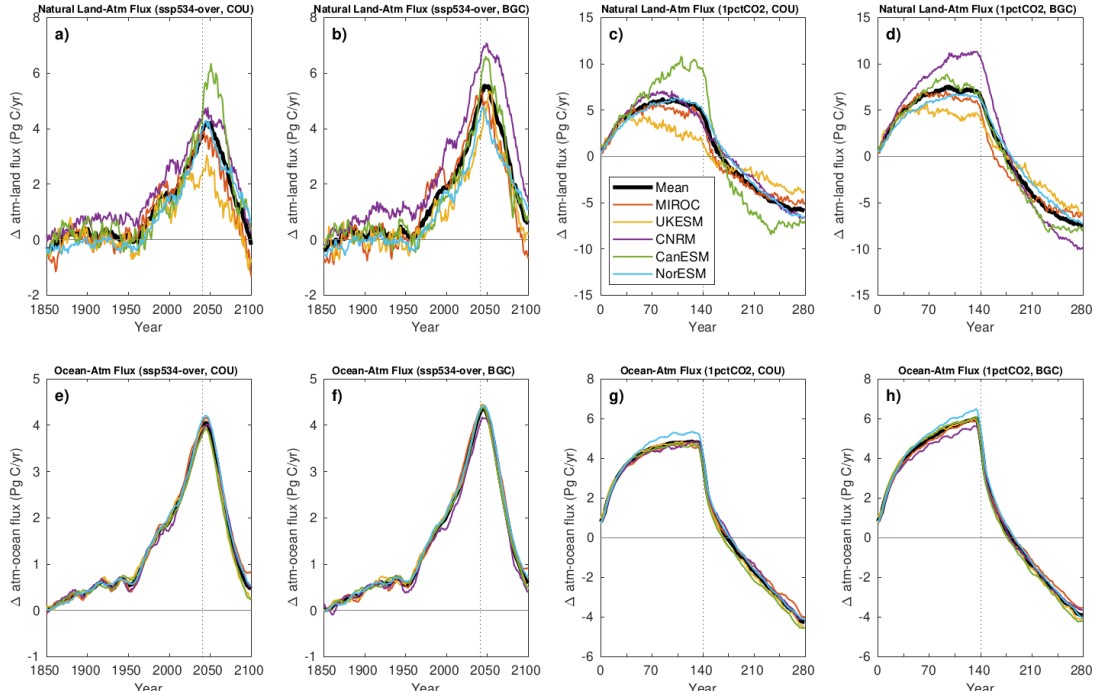

**Figure 2:** Time series of annual mean natural atmosphere-land (a-d) and atmosphere-ocean (e-h) carbon fluxes for the fully and biogeochemically coupled SSP5-3.4-OS and 1pctCO$_2$ experiments as indicated in the panel titles. The dotted vertical lines indicate where [CO$_2$] growth rate peaks in each experiment. An 11-year moving average has been used in all panels.

## 3.2 Global mean carbon cycle feedbacks

### 3.2.1 Ocean

In the BGC simulation, where the effect of changing atmospheric CO$_2$ concentration on terrestrial and marine carbon uptake (the carbon-concentration feedback) is isolated, cumulative atmosphere-ocean carbon fluxes indicate an almost linear growth with [CO$_2$] as long as atmospheric CO$_2$ concentrations are increasing in both SSP5-3.4-OS and 1pctCO2 simulations (Fig. 3a-c). When [CO$_2$] starts to decline, the atmosphere-ocean carbon flux in the 1pctCO2 simulation shows pronounced hysteresis with a continued ocean carbon uptake (until the [CO$_2$]-anomaly has been roughly reduced to 500 ppm) before starting to decrease towards the end of the ramp-down phase (Fig. 3b). In the SSP5-3.4-OS BGC scenario, where the onset of net negative emissions is more gradual, the relationship between cumulative atmosphere-ocean fluxes and [CO$_2$] during the phase of declining atmospheric CO$_2$ concentration also shows hysteresis; but due to the relative short period of net-negative emissions, the ocean remains a sink of carbon in all models until the end of the simulation.



Differences in the cumulative atmosphere-ocean $CO_2$ flux between the COU and the BGC simulations
versus surface temperature changes (carbon-climate feedback) are shown in Fig. 3d-f. Increasing
temperature results in less carbon uptake by the ocean, except for the CNRM-ESM2-1 which simulates
slightly more uptake in the first half of the warming period under the SSP5-3.4-OS. During the negative
emission phases of the simulations when the air surface temperature is decreasing, the carbon-climate
feedback still decreases the ocean carbon content, albeit at reduced rates. Even when pre-industrial
$CO_2$ concentrations are restored at the end of the 1pctCO2 simulation all models agree that the ocean
is still losing carbon due to the effect of (legacy) warming (Fig. 3e). Using the global average ocean
potential temperature (averaged over the full ocean depth) instead of the surface air temperature as a
proxy for oceanic climate change as proposed by Schwinger and Tjiputra (2018), gives a much more
linear relationship between changes in the ocean carbon stock and changes in temperature in the
majority of models (Fig. 3 g-i). At the end of the simulations, the ocean still holds a large part of the
carbon taken up from the atmosphere since pre-industrial time, between roughly 300-400 PgC in
1pctCO2, and around 350 PgC in SSP5-3.4-OS (Fig. S2).
Generally, the ocean carbon-concentration feedback is larger in the SSP5-3.4-OS scenario, which can
most likely be explained with the slower growth rate of [$CO_2$] in this scenario compared to the 1pctCO2
simulation (Fig. 3c). For slower growth rates, the ocean has more time to mix and partly transport the
adsorbed anthropogenic carbon away from the ocean surface to the interior, increasing the capacity
for more uptake. A larger carbon uptake at slower $CO_2$ growth rates has already been reported by
Gregory et al. 2009 and Hajima et al. 2014, although only for combined land and ocean fluxes or land
fluxes only. The ocean carbon-climate feedback, in contrast, is slightly smaller in the SSP5-3.4-OS
scenario, i.e., the carbon loss for a given warming is smaller.



**Figure 3:** Ocean carbon cycle feedbacks in the SSP5-3.4-OS (left column) and 1pctCO2 (middle column) simulations for individual models. The model means for both simulations are shown in the right column. Global mean ocean potential temperature is used on the x-axis of panels (g-i). An 11-year moving average has been used in all panels.

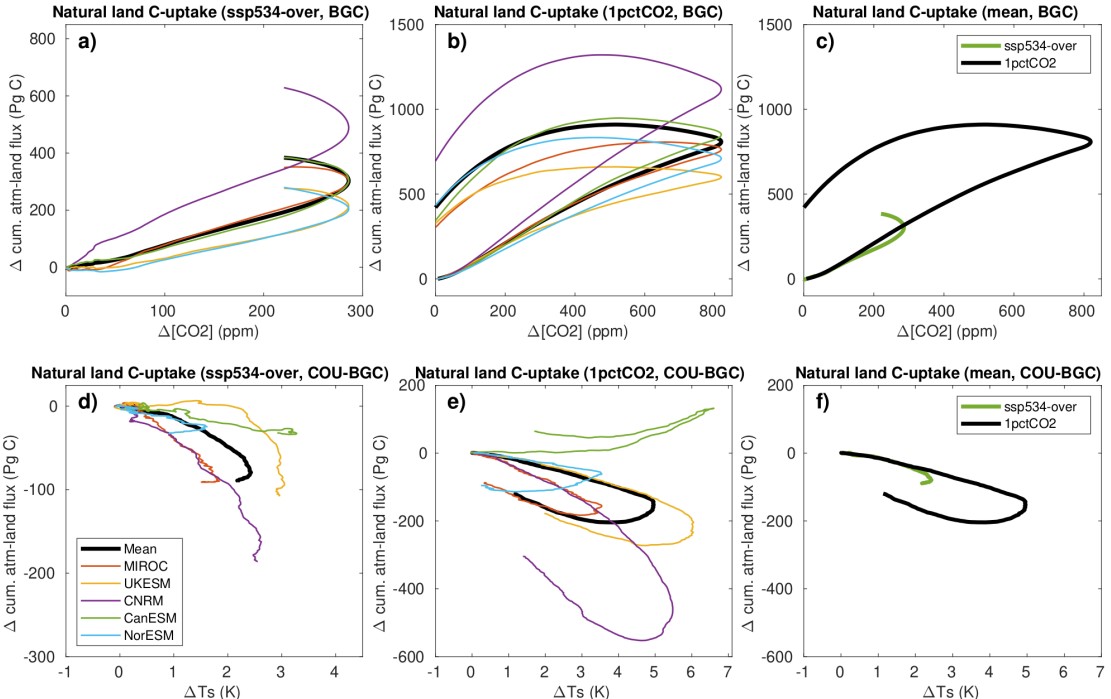

**Figure 4:** Terrestrial carbon cycle feedbacks in the SSP5-3.4-OS (left column) and 1pctCO2 (middle column) simulations for grid cells that are dominated by "natural land" (less than a maximum of 25% crop fraction over the period 2015-2100 in SSP5-3.4-OS). Note that we consider the same grid cells in the 1pctCO2 simulation, even though land use stays at pre-industrial state. The model means for both simulations are shown in the right column. An 11-year moving average has been used in all panels.

### 3.2.2 Land

For grid cells representing natural land, the response of the cumulative terrestrial carbon flux to changes in [$CO_2$] and surface temperature (Fig. 4) is qualitatively similar to the response of the atmosphere-ocean fluxes. In both SSP5-3.4-OS and 1pctCO2 simulations, a roughly linear relationship can be seen between the carbon flux change and both the changes in [$CO_2$] and surface air temperature during positive emission phases. An exception is the carbon-climate feedback of the CanESM5 model, which is about zero up to 4 degrees of warming, and becomes positive for higher temperature increases. This unique behavior is caused by CanESM5's high climate sensitivity combined with larger carbon use efficiency amongst CMIP6 models (as shown later) which causes high latitude vegetation to take up large amounts of carbon in response to warming. This more than compensates for the carbon loss elsewhere associated with climate warming. During negative emission phases both feedbacks show a considerable hysteresis behavior, as for the ocean (see also below).

The carbon-concentration feedback is slightly smaller for the SSP5-3.4-OS scenario compared to the 1pctCO2 experiment (see Fig. 4c), but this difference might be attributed to the remaining influence of land-use changes. This is because, for "crop-land grid cells" (maximum crop-fraction of more than 25%





in the SSP5-3.4-OS scenario), the cumulative carbon fluxes are markedly smaller in the SSP5-3.4-OS
scenario compared to the 1pctCO2 simulation (compare panel c on Figs. S3 and 4). This indicates,
consistent with the results of Melnikova et al. (2022), that the prescribed land use change in the SSP
scenario is the driver behind the small (negative for NorESM2-LM and UKESM) carbon accumulation for
crop land grid cells. We note that land use change is not a feedback process, and it obviously does not
depend on atmospheric $CO_2$ concentration. It is only due to the simulation design used here (see Section
2.2 for details), that the carbon release (or uptake) due to land use changes modifies the net
atmosphere-land $CO_2$ flux which is then seen as a carbon-concentration feedback in the SSP5-3.4-OS-
BGC simulation.
The model-mean carbon-climate feedback for natural land is very similar for the SSP5-3.4-OS and
1pctCO2 simulations during the positive emission phases, but deviates thereafter due to hysteresis
behavior (Fig. 4f). Interestingly, in contrast to the carbon-concentration feedback, the global average
carbon-climate feedback for cropland and natural land remains very similar between the SSP5-3.4-OS
and 1pctCO2 simulations (Fig. S3). This is likely due to the similar response of the soil carbon to changes
in surface air temperature.

### 3.2.3 Hysteresis

For the 1pctCO2 simulation, hysteresis can be defined as the difference in, for example, cumulative
carbon uptake during the ramp-up and the ramp-down period at the same level of atmospheric $CO_2$
concentration. Here, to quantify hysteresis, we choose the years 70 and 210, which represent a state
where atmospheric $CO_2$ has been doubled (570 ppm) or returned to this value after the overshoot. We
refrain from quantifying hysteresis for the SSP5-3.4-OS scenario, because of the relatively short period
of declining [$CO_2$].
The model mean hysteresis in the carbon-concentration feedback is 443±29 PgC (model uncertainty
measured as one standard deviation) for the ocean and 524±205 PgC for natural land, which for both
cases is larger than the feedback at year 70 itself. Although the hysteresis of the ocean carbon-
concentration feedback is smaller than the terrestrial feedback in absolute terms, it is larger in relative
terms (179% of the accumulated carbon uptake at year 70 for the ocean versus 168% for land). In
general, the hysteresis seems to be related to the magnitude of the carbon-concentration feedback,
since models with a large (small) carbon uptake at year 140, tend to show a large (small) hysteresis at
year 210 for both ocean and land. However, towards the end of the ramp-down period, this relationship
breaks down for CanESM5 and MIROC, particularly over land.
For the carbon-climate feedback, the hysteresis in climate induced carbon loss or gain (difference
between COU-BGC evaluated at years 70 and 210) is -102±22 and -158±181 PgC for ocean and natural
land, respectively. As for the carbon-concentration effect, a relationship between the magnitude of
carbon loss or gain at year 140 and the hysteresis is found. Models with a large (small) climate induced
loss of carbon tend to have a large (small) hysteresis.






### 3.3 Carbon cycle feedback metrics

#### 3.3.1 Model mean global land and ocean responses

We now discuss the model-mean time evolution of the feedback metrics $\beta$ and $\gamma$ (Eqs. 1 and 2) derived from the 1pctCO2 and SSP5-3.4-OS simulations. In the SSP5-3.4-OS scenario (Fig. 5a) the model-mean feedback metric $\beta_L$ increases monotonically from about 0.7 to 1.9 PgC ppm$^{-1}$ during the period 2000-2100. Over the ocean, $\beta_O$ in the SSP5-3.4-OS scenario decreases slightly until the mid-21st century, and then it rises to about 1.7 PgC ppm$^{-1}$. Due to the much larger spread in carbon fluxes over land (Fig. 2), the resulting model spread for both $\beta_L$ and $\gamma_L$ is also much larger than for $\beta_O$ and $\gamma_O$.

For the 1pctCO2 simulation, during the ramp-up phase over both land and ocean (Fig. 5b), $\beta$ initially increases and then decreases slightly with increasing [CO$_2$] consistent with the results of Arora et al. (2013) for the same experiment but using CMIP5 ESMs. In contrast, during the ramp-down phase of the 1pctCO2-cdr experiment, $\beta$ reaches very high values over both land and ocean (Fig. 5c). This is because, during the carbon removal phase of the 1pctCO2-cdr experiment, there is a much larger amount of accumulated ocean and terrestrial carbon for the same atmospheric CO$_2$ concentration due to the large hysteresis seen in Figs. 3 and 4. Eventually, while [CO$_2$] is approaching pre-industrial values (i.e., Δ[CO$_2$] reaches zero), changes in cumulative fluxes (i.e., carbon stocks) relative to their pre-industrial values remain positive, making $\beta$ ill-defined towards the end of the 1pctCO2 ramp-down. For the same reason, an increase of $\beta_L$ and $\beta_O$ is also seen in the SSP5-3.4-OS scenario after the CO$_2$ concentration peak in 2062.

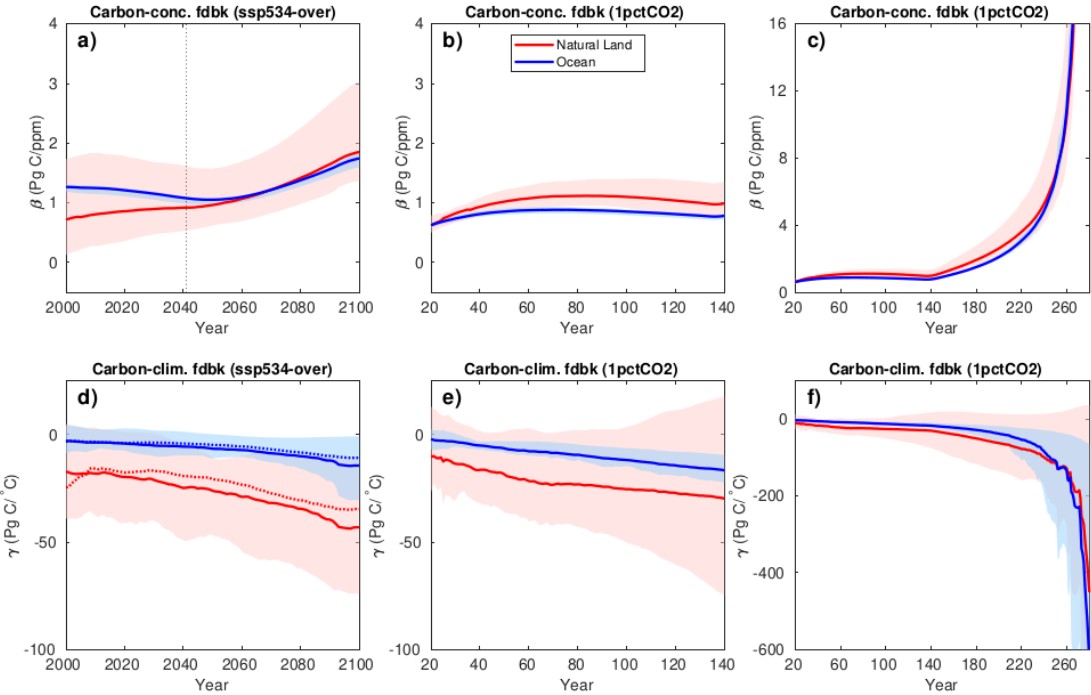




**Figure 5:** Model-mean $\beta$ (a-c) and $\gamma$ (d-f) feedback metrics in the SSP5-3.4-OS and 1pctCO2 experiments
for natural land and ocean. Panels (b and e) show a zoom into the ramp-up phase of the time series
shown on panels (c and f). Shadings show the range across the models. The dotted vertical line on panel
a indicates where [$CO_2$] growth rate peaks in the fully coupled SSP5-3.4-OS experiment. Dotted curves
on panel d indicate the model mean with the assumption of negligible temperature change in the BGC
simulation. An 11-year moving average has been used in all panels.

The model mean feedback factor $\gamma$ is negative as the impact of climate change generally reduces the
carbon stocks of land and ocean. In both SSP5-3.4-OS and 1pctCO2 experiments, the carbon-climate
feedback is increasing over time (more negative $\gamma$, Fig. 5d and e), similar to figure 6 of Arora et al.
(2013). The carbon-climate feedback is generally much smaller for the ocean than for land, and the
model uncertainty for $\gamma_O$ is only a small fraction of $\gamma_L$. Note that the same globally averaged surface air
temperature anomaly is being used for the calculation of both $\gamma_O$ and $\gamma_L$ (Eq. 2). As noted above, the
CanESM5 model simulates a globally increasing land uptake due to climate change towards the end of
the 1pctCO2 simulation (Fig. 4e), resulting in a positive $\gamma_L$ for this model. During the ramp-down phase
of the 1pctCO2 experiment (Fig. 5f), $\gamma$ reaches very large negative values. Similar to $\beta$, this is caused by
the large hysteresis of the climate change impact on cumulative carbon stock while the surface
temperature change becomes small (see Eq. 2). The assumption of $\Delta T^{BGC} = 0$ generally works well
except for $\gamma_L$ in the SSP5-3.4-OS scenario where non-$CO_2$ forcings have a significant contribution to
$\Delta T^{BGC}$ (dashed curves in Fig. 5d).
The global feedback factors B and Γ for the SSP5-3.4-OS and 1pctCO2 simulations are shown in Fig. S4.
This feedback metric directly reflects the instantaneous fluxes, not cumulative fluxes, and is therefore
less influenced by the history of carbon fluxes, unlike $\beta$ and $\gamma$. Consistent with Fig. 2, the model-mean
B remains positive during the SSP5-3.4-OS simulation and during the positive emission phase of the
1pctCO2 both over natural land and ocean. Only one model indicates a negative carbon-concentration
feedback over natural land towards the very end of the SSP5-3.4-OS simulation during its relatively
short negative emission phase. B reflects the saturation of carbon sinks in the 1pctCO2 simulation with
time and decreases monotonically during the positive emission phase. Similar to what we have seen
earlier for $\beta$, B shows large but negative values towards the end of the 1pctCO2 ramp-down phase (Fig.
S4c).
An interesting difference between the $\gamma$ and Γ feedback metrics is seen towards the end of the 1pctCO2
negative emissions phase (Fig. S4f), where $\Gamma_L$ turns positive around year 180. This indicates that the land
biosphere starts gaining carbon that was previously lost due to the impacts of climate change. In
contrast, $\Gamma_O$ remains negative indicating that the ocean continues to lose carbon due to warmer than
pre-industrial conditions until the end of the 1pctCO2 ramp-down phase. Because they are based on
cumulative emissions, both $\gamma_O$ and $\gamma_L$ remain negative throughout the 1pctCO2 ramp-down. This
illustrates that the use of a feedback metric based on time-integrated carbon fluxes might obscure
changes in important processes during net-negative emission phases. Eventually, both approaches for
calculating feedback metrics become ill-defined when the deviation of [$CO_2$] or temperature from their
pre-industrial values becomes small. This implies that both feedback metrics are not suited to describe





feedbacks towards the end (and beginning) of a concentration driven simulation set-up where pre-
industrial concentrations are restored.

**3.3.2 Model uncertainties and relative feedback strength in global feedback metrics**
Figure 6 shows the model spread of feedback metrics at different points in time for the 1pctCO2
simulation and the SSP5-3.4-OS scenario (see also Table 2). The larger model-mean values during the
negative emission phases have been discussed in the previous section, but Fig. 6 also shows a strong
increase in model uncertainty (measured as the standard deviation around the model mean, Table 2)
between the ramp-up and ramp-down phase of the 1pctCO2 simulation. For both $\beta_L$ and $\beta_O$, there is
either no ($\beta_O$) or only a small ($\beta_L$) increase in model uncertainty between the years 70 and 140 of the
1pctCO2 simulation, whereas at year 210 uncertainty has increased by about a factor of four. This
"jump" in uncertainty in $\beta$ is solely caused by differences in how the models react to the sharp change
in forcing from increasing to decreasing $CO_2$ at year 140 (see Eq. 1, note that atmospheric $CO_2$ is
prescribed and $\Delta T^{BGC}$ is small). A similar behavior is seen for $\gamma_O$, while for $\gamma_L$ the increase in model
uncertainty is more gradual, i.e., the increase between years 70 and 140 is about the same as between
years 140 and 210. There is also a consistent increase in model uncertainty in all feedback metrics from
the positive to the negative emissions phase in the SSP5-3.4-OS scenario.
The relative strength of the feedback among the models remains relatively stable over time, between
positive and negative emission phases, and between the different experiments. Model A having a
stronger (weaker) feedback than model B at one of the instances depicted in Fig. 6, indicates that model
A will have a stronger (weaker) feedback than model B for the other instances with only few exceptions.
Most of these exceptions arise because modeled feedbacks are very similar such that small changes in
feedback strength can lead to a different ranking. In a few cases relative feedback strength evolves
differently in the models. For example, NorESM2-LM evolves from having a weaker than average $\gamma_L$ in
the positive emission phase of the 1pctCO2 simulation to having a stronger than average $\gamma_L$ in the
negative emission phase.
Finally, it is worth noting that while the model uncertainty in $\gamma_O$ is much smaller than in $\gamma_L$ during the
ramp-up phase of the 1pctCO2 simulation (uncertainty in $\gamma_O$ is only 15% of those in $\gamma_L$ at year 140), this
situation has changed for the ramp-down phase. At year 210, the uncertainty in the ocean carbon-
climate feedbacks has grown much stronger than the uncertainties of the terrestrial carbon-climate
feedback, such that model uncertainties in $\gamma_O$ are 42% of those in $\gamma_L$.




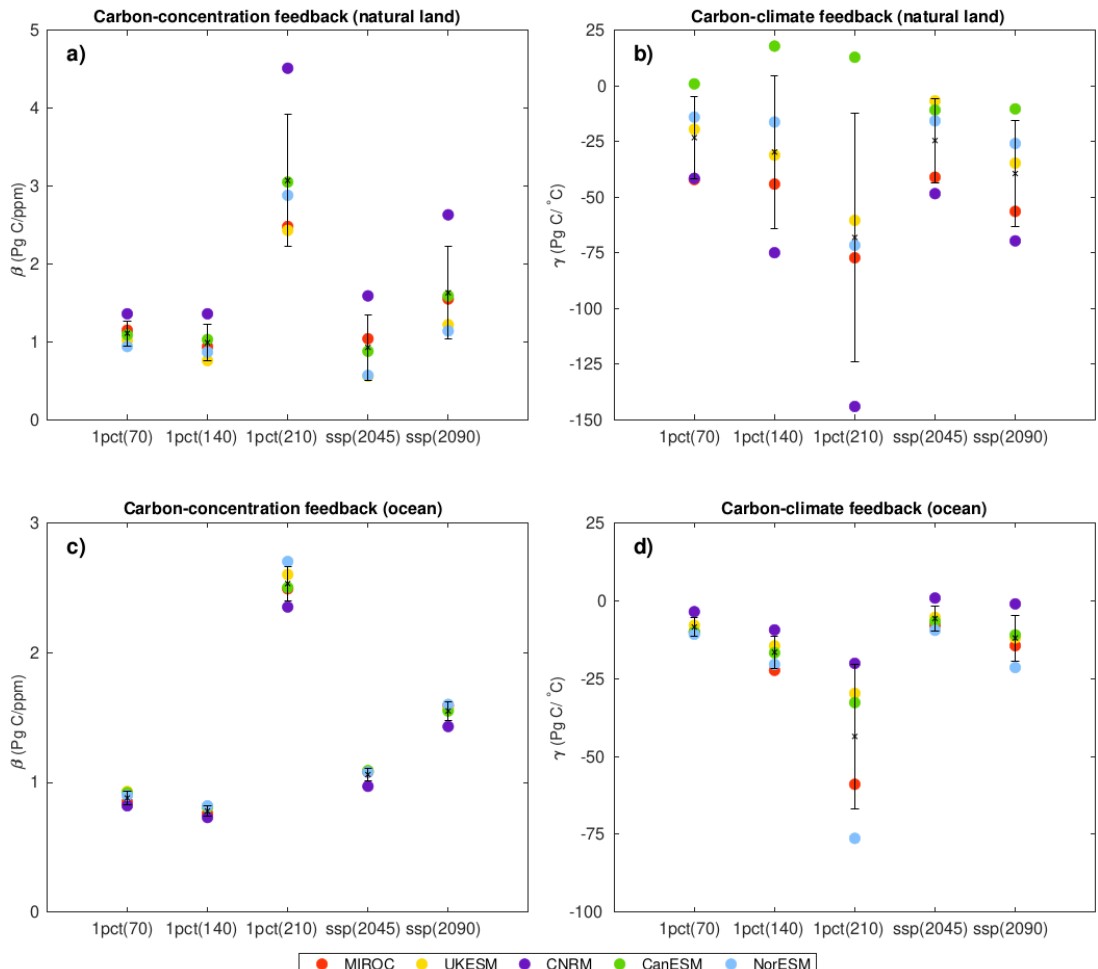


**Figure 6:** Globally averaged values of $\beta$ (a and c) and $\gamma$ (b and d) feedback metrics in the 1pctCO2 (years
70, 140, and 210) and SSP5-3.4-OS (years 2045 and 2090) experiments for natural land and ocean. The
bars show the mean ± 1 standard deviation range, and the individual colored dots represent individual
models.

**Table 2:** Globally averaged values of $\beta$ (Pg C ppm$^{-1}$) and $\gamma$ (Pg C °C$^{-1}$) feedback metrics at years 70, 140, and 210
of the 1pctCO2 simulation and years 2045 and 2090 at the ramp-up and ramp-down phases of the SSP5-3.4-OS
experiment for natural land and ocean.

|  | MIROC-ES2L | UKESM1-0-LL | CNRM-ESM2-1 | CanESM5 | NorESM2-LM | Mean |
|---|---|---|---|---|---|---|
| $\beta_{L(70)}$ | 1.15 | 1.02 | 1.36 | 1.09 | 0.94 | 1.11 (SD=0.16) |



| | | | | | |
|---|---|---|---|---|---|
| $\beta_{L(140)}$ | 0.94 | 0.76 | 1.36 | 1.03 | 0.87 | 0.99 (SD=0.23) |
| $\beta_{L(210)}$ | 2.48 | 2.43 | 4.51 | 3.05 | 2.88 | 3.07 (SD=0.85) |
| $\beta_{L(2045)}$ | 1.04 | 0.56 | 1.59 | 0.88 | 0.57 | 0.93 (SD=0.42) |
| $\beta_{L(2090)}$ | 1.55 | 1.22 | 2.63 | 1.59 | 1.14 | 1.63 (SD=0.59) |
| $\gamma_{L(70)}$ | -42.14 | -19.54 | -41.58 | 0.82 | -14.12 | -23.31 (SD=18.5) |
| $\gamma_{L(140)}$ | -44.17 | -31.19 | -74.97 | 17.78 | -16.31 | -29.77 (SD=34.3) |
| $\gamma_{L(210)}$ | -77.26 | -60.45 | -144.01 | 12.77 | -71.64 | -68.12 (SD=55.8) |
| $\gamma_{L(2045)}$ | -41.08 | -6.80 | -48.46 | -10.93 | -15.78 | -24.61 (SD=18.9) |
| $\gamma_{L(2090)}$ | -56.43 | -34.76 | -69.66 | -10.41 | -25.95 | -39.44 (SD=23.7) |
| $\beta_{O(70)}$ | 0.85 | 0.93 | 0.82 | 0.92 | 0.90 | 0.88 (SD=0.05) |
| $\beta_{O(140)}$ | 0.76 | 0.81 | 0.73 | 0.81 | 0.82 | 0.78 (SD=0.04) |
| $\beta_{O(210)}$ | 2.49 | 2.60 | 2.35 | 2.50 | 2.70 | 2.53 (SD=0.13) |
| $\beta_{O(2045)}$ | 1.08 | 1.09 | 0.97 | 1.09 | 1.08 | 1.06 (SD=0.05) |
| $\beta_{O(2090)}$ | 1.59 | 1.57 | 1.43 | 1.55 | 1.60 | 1.55 (SD=0.07) |
| $\gamma_{O(70)}$ | -10.09 | -7.95 | -3.60 | -10.13 | -10.84 | -8.52 (SD=2.96) |
| $\gamma_{O(140)}$ | -22.40 | -14.56 | -9.44 | -16.77 | -20.48 | -16.61 (SD=5.10) |
| $\gamma_{O(210)}$ | -58.94 | -29.78 | -20.16 | -32.75 | -76.28 | -43.59 (SD=23.2) |
| $\gamma_{O(2045)}$ | -7.88 | -5.43 | 0.78 | -6.75 | -9.56 | -5.77 (SD=3.96) |
| $\gamma_{O(2090)}$ | -14.50 | -11.98 | -1.10 | -11.05 | -21.50 | -12.03 (SD=7.35) |



### 3.3.3 Model differences in the terrestrial carbon-concentration feedback

Figure 7 shows the individual components of the decomposition of $\beta$ (Eq. 3), separately for tropical and
subtropical (30°S-30°N) and higher latitudes (between 30°N/S and poles), both on the ramp-up and
ramp-down phases (years 70 and 210, respectively) of the 1pctCO2-bgc experiment. The time periods
are selected such that the atmospheric $CO_2$ concentration is the same (569 ppm, a doubling of pre-



industrial $CO_2$ concentration). All models consistently show increases in both $\tau_{cveg\Delta}$ and $\tau_{csoil\Delta}$ during
the ramp-down compared to the ramp-up phase, since these metrics are based on cumulative
vegetation and soil carbon (Eq. 3), which are slower than NPP and GPP in reacting to decreasing $[CO_2]$.
Lower (higher) latitudes are associated with higher $\tau_{cveg\Delta}$ ($\tau_{csoil\Delta}$). Likewise, the litterfall term $\frac{\Delta LF}{[CO_2]}$ is
larger during the ramp-down phase in all models due to lagged reaction of vegetation carbon to the
decrease in $[CO_2]$, with this effect being generally most pronounced at low latitudes. There is also a
consistent but small increase in the term $\frac{\Delta GPP}{[CO_2]}$ , which represents the $CO_2$ fertilization effect. This
increase implicitly includes the effect of changes (typically an increase) in standing vegetation biomass
and leaf area index for all models but also changes in vegetation cover as $[CO_2]$ varies for UKESM that
simulates dynamic vegetation cover. For the dimensionless fractions $\frac{\Delta R_h}{\Delta LF}$ and $CUE_\Delta$, changes between
ramp-up and ramp-down phases are less consistent between the models. For $CUE_\Delta$, three models show
an increase and two models a decrease, although the changes between ramp-up and ramp-down
phases are generally small. For $\frac{\Delta R_h}{\Delta LF}$ changes range from a 115% increase (CNRM at low latitudes) to a
small decrease (UKESM). It is worth noting that for four out of six terms of Eq. 3 ($\tau_{cveg\Delta}$, $\tau_{csoil\Delta}$, $\frac{\Delta R_h}{\Delta LF}$,
and $\frac{\Delta LF}{[CO_2]}$) the model disagreement is significantly larger during the ramp-down phase of the 1pctCO2
simulation, indicating that changes in these processes are responsible for the strong increase in model
uncertainty in $\beta_L$ between positive and negative emission phases pointed out in the previous section.
The decomposition applied here helps to understand some of the model differences visible in Fig. 4. As
already pointed out in Arora et al. (2020), the high accumulation of terrestrial carbon by the CNRM-
ESM2-1 model in the BGC simulation (Fig. 4b) is not caused by a particularly strong $CO_2$ fertilization
effect or $CUE_\Delta$ but rather by relatively high values of $\tau_{cveg\Delta}$ and $\tau_{csoil\Delta}$, indicating long residence
timescales in vegetation and soil. Likewise, CanESM5's higher than average atmosphere-land C flux (Fig.
4b), despite its near-average strength of the $CO_2$ fertilization effect and soil and vegetation turnover
times is due to its high $CO_2$ fertilization effect at lower latitudes and also its high $CUE_\Delta$ through which
the model converts a much larger fraction of GPP to NPP. Compared to the other models, CanESM5
also shows the largest relative increase (85% and 134% for lower and higher latitudes, respectively) in
$\tau_{csoil\Delta}$ between years 70 and 210.



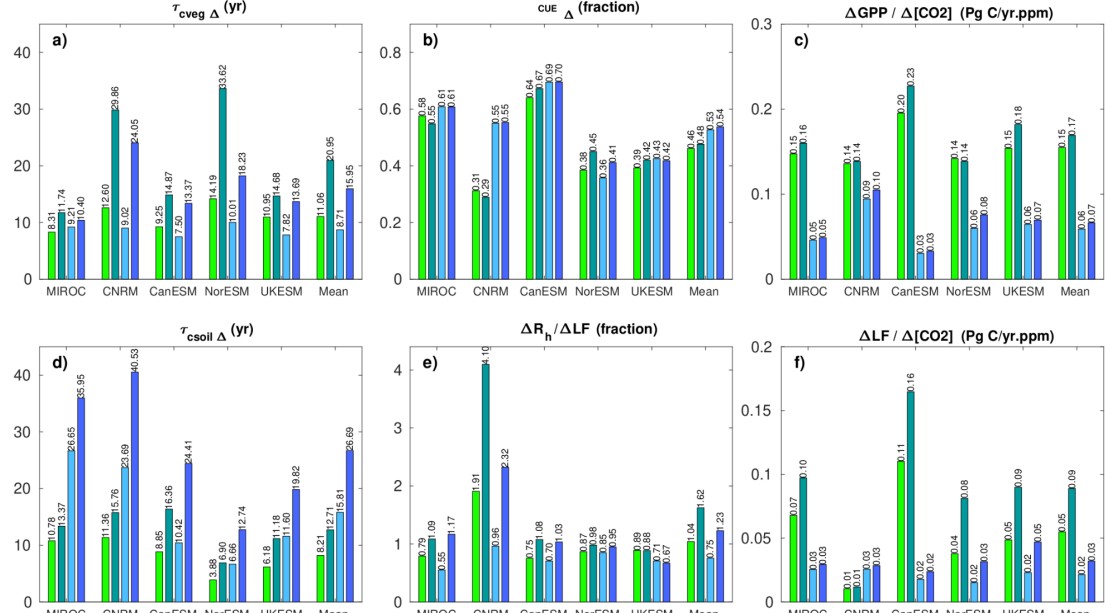

**Figure 7:** Individual terms of Eq. (3) contributing to $\beta_L$. Values for tropical and subtropical (between 30°S and 30°N) regions are in green, and for northern latitudes (above 30°S and 30°N) are in blue. Lighter (darker) color on each panel corresponds to the middle of the ramp-up (ramp-down) phase of the 1pctCO2-bgc and 1pctCO2-cdr-bgc experiments (years 70 and 210, respectively).

### 3.3.4 Northern hemisphere high-latitude permafrost and non-permafrost regions

Of the models considered here, only NorESM2-LM has a terrestrial model that vertically resolves soil carbon (CLM5, Lawrence et al. 2019). Since this is a prerequisite to skillfully simulate carbon release during gradual permafrost degradation, we restrict our analysis of high latitude and permafrost feedbacks to the NorESM2-LM model. If only natural land is considered, the area associated with permafrost and non-permafrost regions north of 45°N is about 14.7 and 17.5 x10⁶ km², respectively (total area is 14.7 and 24.1 x10⁶ km²).

The effect of warming on carbon uptake in the high-latitude non-permafrost region is positive ($\gamma > 0$, increased uptake) in NorESM2-LM in both the SSP5-3.4-OS and 1pctCO2 simulation (Fig. 8a-c, blue lines). Within the permafrost region, $\gamma$ is close to zero for the SSP5-3.4-OS simulation up to 2100 and the ramp-up phase of the 1pctCO2 simulation (Fig. 8a,b, red line), albeit with a decreasing (more negative) trend. This is due to a compensation of vegetation carbon gain and soil carbon losses (Fig. S5). During the ramp-down phase of the 1pctCO2 simulation, permafrost soil carbon losses increase approximately until year 210 of the simulation (Fig. S5). Thereafter, permafrost soil carbon stays roughly constant with a cumulative loss of about 55 PgC over the simulation. Vegetation carbon over the permafrost region still increases for the first 30 years of the ramp-down phase of the 1pctCO2




simulation, after which it decreases mainly due to decreasing temperature (Fig. S5g). The $\gamma$ value
calculated for the permafrost region, therefore, shows a sharp decrease during the ramp-down period
of the 1pctCO2 simulation (Fig. 8c). Eventually, when $\Delta T$ approaches small values $\gamma$ loses its significance
as seen before for the global feedback factors.

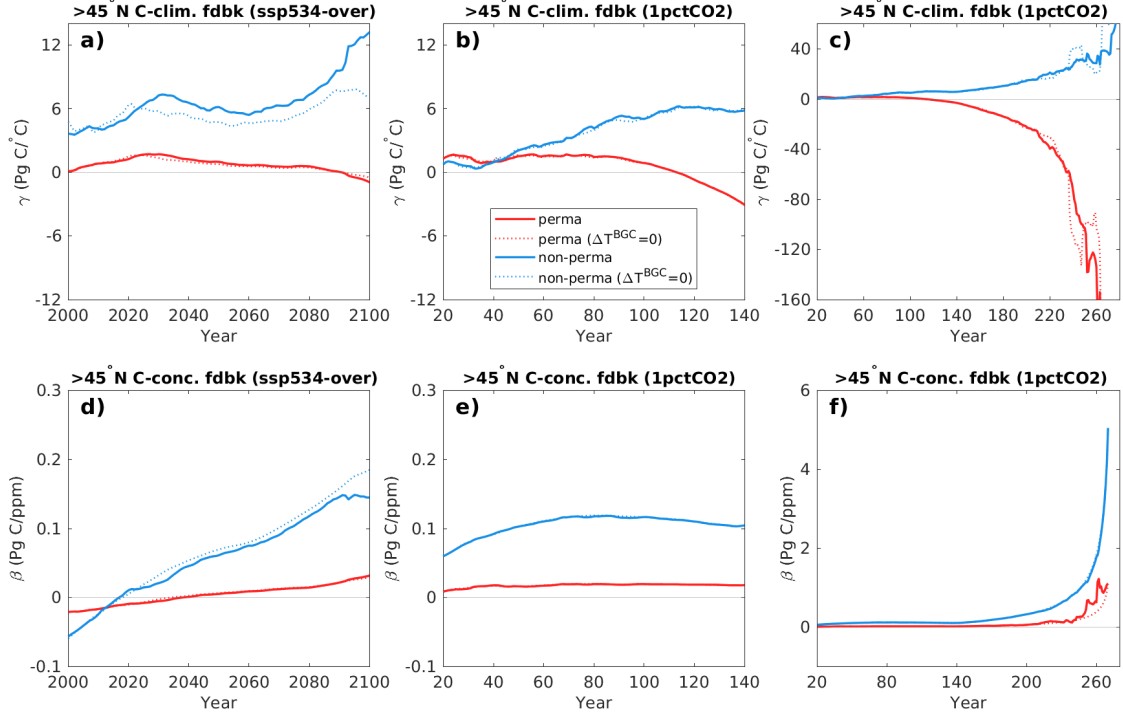


**Figure 8:** $\gamma$ (a-c) and $\beta$ (d-f) for northern hemisphere high latitude natural land permafrost and non-
permafrost regions in the SSP5-3.4-OS and 1pctCO2 simulations using the NorESM model. An 11-year
moving average has been used in all panels.

In both the SSP5-3.4-OS scenario and the 1pctCO2 simulations, $\beta$ is positive (except initially in the SSP5-
3.4-OS simulation) although the absolute values remain very small. The carbon-concentration feedback
is stronger over the non-permafrost area, where both soil and vegetation carbon increase in the BGC
simulation, than over the permafrost area, where soil and vegetation carbon stay almost constant in
BGC (Fig. S5).
NorESM2-LM has the smallest transient climate response (TCR) of the models considered here, and it
can be expected that the permafrost carbon-climate feedback estimated here would be larger in a
model with higher TCR. Nevertheless, the permafrost carbon loss of 26.9 Pg C °C$^{-1}$ in the year 210 of the
simulation contributes 38% of the total carbon-climate feedback at this point in time in NortESM2-LM.



### 3.4 Geographical pattern of carbon cycle feedback metrics

We have calculated $\beta$ and $\gamma$ feedback factors at grid-scale to assess the spatial patterns of feedbacks over the land and ocean (Figs. 9 and 10). In order to compare positive and negative emission phases, we selected 21-year time intervals centered at years 70 and 210 of the ramp-up and ramp-down phases of the 1pctCO2 simulation, at an atmospheric $CO_2$ concentration of 570 ppm (corresponding to a doubling of pre-industrial $CO_2$ concentration). We also selected a 21-year time-interval centered at year 2045 (corresponding to $CO_2$ concentration of 523 ppm), shortly before the $CO_2$ peak of the SSP5-3.4-OS scenario. We have also analyzed a 21-year time interval during the net-negative emission phase of the SSP scenario (centered at year 2090), but since the time-period of net-negative emissions in the SSP-scenario is relatively short, we focus on comparing the feedbacks during the positive and negative emission phases of the 1pctCO2 simulation alongside with the feedbacks during the positive emission phase of SSP5-3.4-OS. For completeness, Fig. S6 shows the spatially resolved feedback during the net-negative emission phase of SSP5-3.4-OS.

In the 1pctCO2 simulation, rising [$CO_2$] increases the modeled carbon sinks almost everywhere (i.e., positive $\beta$) over the land and ocean (Fig. 9a-e). CanESM5 shows a weak negative $\beta$ over northern high-latitude land areas, and there are some spurious negative values of $\beta$ over desert areas in some models. For the ocean, all models agree that the regions with the strongest increase of the oceanic $CO_2$ sinks in response to higher [$CO_2$] are the North Atlantic and the Southern Ocean. As seen for the global average (Fig. 5), $\beta$ remains positive and increases in magnitude during the ramp-down phase (Fig. 9 f-j, note the different color scale). As an overarching observation, the large scale patterns of the carbon-concentration feedback are remarkably similar during the ramp-up and ramp-down phases of the 1pctCO2 simulation (with spatial correlations, averaged across all the models, of 0.93 and 0.80 over land and the ocean, respectively) but the magnitude of the feedback is about two times larger in the ramp-down phase, consistent with the lagged response of cumulative carbon uptake to the decrease in atmospheric $CO_2$ (Figs. 3 and 4). The most prominent change in the spatial pattern of $\beta$ occurs in the equatorial Pacific. All models consistently show that this area has turned from a cumulative carbon sink at year 70 to a cumulative carbon source at year 210.

We find the largest values of $\beta$ over tropical land and to a lesser extent over northern hemisphere temperate and boreal ecosystems coincident with areas of large biomass (forests). For three of the models (NorESM2-LM, CanESM5, and UKESM1-0-LL), the feedback is clearly dominated by tropical and subtropical regions, while for MIROC-ES2L the feedback is approximately of the same strength in northern temperate and high-latitude regions. For CNRM-ESM2-1, the carbon-concentration feedback is on average stronger north of 30° latitude than in tropical/subtropical regions. For NorESM2-LM and UKESM1-0-LL, the tropical dominance of the carbon-concentration feedback stems from vegetation carbon, while for CanESM5 both vegetation and soil carbon contribute about equally (Figs. S7 and S8).

The results presented in Section 3.3.3 provide to some extent a mechanistic understanding of these model differences. CNRM-ESM2-1 has the highest $CO_2$ fertilization effect $\frac{\Delta GPP}{[CO_2]}$ in high latitudes and



the lowest CUEΔ at low latitudes. This, combined with a large high-latitude $\tau_{csoil\Delta}$ leads to a larger
carbon accumulation in vegetation and soil in higher latitudes than in the tropics/subtropics in this
model. The three models with tropical dominance of $\beta$ (NorESM2-LM, CanESM5, and UKESM1-0-LL)
have a relatively high $\tau_{cveg\Delta}$ and relatively low $\tau_{csoil\Delta}$. CanESM5, shows the strongest
tropical/subtropical $CO_2$ fertilization effect, but also a large response of the litterfall term leading to
large responses in both vegetation and soil carbon.
In the SSP5-3.4-OS simulation, the ocean $\beta$ magnitude is similar to that of the 1pctCO2 simulation and
the spatial distribution of the ocean response to the [$CO_2$] rise is roughly consistent between the models
(Fig. 9k-o). In contrast, the feedback pattern over natural land is different in some regions and models
between the SSP scenario simulation and the idealized 1pctCO2 experiment. UKESM1-0-LL, CanESM5,
and to a lesser extent NorESM2-LM project negative $\beta$ values in some northern high latitude regions
(e.g., Siberia). These negative $\beta$ values are either not seen at all (UKESM1-0-LL, NorESM2-LM) or are
weaker (CanESM5) in the 1pctCO2 simulation, and they originate from a combination of vegetation and
soil carbon pools (Figs. S7 and S8). Unlike in the 1pctCO2 experiment, temperature changes are not
negligible in the BGC simulation of the SSP5-3.4-OS experiment (Fig. 1). Nevertheless, the spatial
distribution of the feedback factor $\beta$ calculated with the assumption $\Delta T^{BGC} = 0$ results in a similar
pattern (not shown), which suggests that the non-negligible temperature changes in the BGC simulation
are not the cause for these negative values of $\beta$. Rather, these negative values are most likely caused
by remaining land use change in grid cells that we have classified as "natural" land with our simple
threshold approach. This is consistent with the fact that the high-latitude negative $\beta$ values occur in
those models that have low $\beta$ in these regions in the 1% simulation (i.e., a relatively small land use
change perturbation can change $\beta$ from positive to negative).



**Figure 9:** The spatial distribution of $\beta$ (kg C m$^{-2}$ ppm$^{-1}$) at year 70 of the ramp-up phase of the 1pctCO2 simulation (a-e), at year 210 of the ramp-down phase of the 1pctCO2 simulation (f-j), and at year 2045 (natural land only, white areas are crop-dominated grid cells) during the positive emission phase of the SSP5-3.4-OS scenario (k-o).

Figure 10 indicates that the ESMs considered here simulate predominantly negative values of $\gamma_O$ over the ocean. Positive values of $\gamma_O$ are found in the Arctic, and in some cases in parts of the polar Southern Ocean adjacent to Antarctica. Climate change increases the ocean $CO_2$ sink in these regions mainly due to a reduction in sea ice coverage (Roy et al. 2011; Schwinger et al. 2014). The North Atlantic Ocean and the Southern Ocean have the largest negative $\gamma_O$ values due to changes in ocean circulation and



deep water formation. In tropical and subtropical ocean regions, the reduced oceanic carbon uptakes
can be attributed to warming-induced decreased $CO_2$ solubility and increased stratification (Roy et al.
707    2011).

Over land, climate change generally reduces carbon sinks in the tropics and mid-latitudes. In the high
latitudes models disagree on the strength and the sign of the carbon-climate feedback. CNRM-ESM2-1
shows relatively strong soil carbon losses in northern high latitudes, which overcome vegetation carbon
gains (Fig. S9 and S10) leading to mostly negative values of $\gamma_L$ in this region. As mentioned above,
CanESM5's carbon-climate feedback switches from weak negative at 2xCO2 to positive at 4xCO2. Figure
9c clearly shows that the positive global $\gamma$ values originate from the northern hemisphere high latitudes.
Also, the positive $\gamma_L$ in CanESM5 over the northern high latitudes is seen in both vegetation and soil
carbon reservoirs, but with a time lag for soil carbon. Consistent with our analysis in Sect. 3.3.4,
NorESM2-LM shows permafrost carbon loss in north-eastern Siberia and northern Alaska, but these
losses become significant only during the ramp-down phase of the 1pctCO2 simulation (Fig. 9j).
The spatial pattern of the carbon-climate feedback is similar during the ramp-up and ramp-down phases
of the 1pctCO2 simulation, but the magnitude has roughly doubled during the ramp-down phase,
consistent with the cumulative nature of the $\gamma$ feedback metric used here (note the different color-
scales in Fig. 9). The correlations of the spatial patterns (at years 70 and 210) are lower than for $\beta$ and
range from 0.41 (MIROC-ES2L) to 0.66 (UKESM1-0-LL) for $\gamma_O$ and from 0.49 (NorESM2-LM) to 0.88
(UKESM1-0-LL) for $\gamma_L$.
The value of the $\gamma$ feedback metric in the SSP5-3.4-OS scenario simulation is less affected by land-use
change, since the same land-use changes are imposed in both the COU and the BGC simulation. In
contrast to $\beta$, which is directly altered by carbon stock changes due to land-use changes, $\gamma$ is only
influenced indirectly, possibly by different sensitivities of the new vegetation cover after a land-use
transition, or by changes in local to regional climatic conditions. In the global mean, the carbon-climate
feedback during the positive emission phase is very similar for the SSP scenario and the 1pctCO2
simulation (Fig. 5d and e). Also, the spatial patterns of $\gamma_L$ are largely similar between the SSP5-3.4-OS
and the ramp-up phase of the 1pctCO2 simulation with correlations ranging from 0.71 (NorESM2-LM)
to 0.84 (CNRM-ESM2-1). The largest difference between the two simulations is an enhanced positive
feedback over northern high-latitude land in the UKESM1-0-LL model in SSP scenario compared to the
1pctCO2 simulation, which is seen in both vegetation and soil carbon pools (Figs. S9 and S10).
Over the ocean the global mean carbon-climate feedback is slightly smaller in SSP5-3.4-OS compared
to the 1pctCO2 simulation (Fig. 3f), but again, the spatial pattern is largely similar with correlations
ranging from 0.47 (CNRM-ESM2-1) to 0.78 (MIROC-ES2L).





**Figure 10:** same as Fig. 9 but for $\gamma$ (kg C m$^{-2}$ °C$^{-1}$). Note that cropland areas are not excluded from panels (k-o) as in Fig. 9.

## 4. Summary and conclusions

We have investigated carbon cycle feedbacks in a highly idealized model experiment with exponentially increasing and decreasing atmospheric $CO_2$ concentration (1pctCO2) and in a more realistic overshoot scenario simulation (SSP5-3.4-OS). We employ an ensemble of five CMIP6 ESMs that have run additional (biogeochemically coupled) simulations that allow us to separate the effects of changing atmospheric $CO_2$ and of changing surface climate on the simulated carbon cycle. We discuss global mean carbon fluxes and employ the widely used carbon cycle feedback metrics of $\beta$ and $\gamma$ (Friedlingstein et al. 2003) to compare feedbacks between models and between phases of (implied) positive and negative $CO_2$





emissions as well as the (model) uncertainty of these feedbacks. To determine the sources of
uncertainty for the terrestrial carbon-concentration feedback, we also decompose $\beta_L$ into contributions
from different processes following the methodology of Arora et al. (2020), and investigate spatial
feedback patterns and their changes.
We find that both the carbon-concentration ($\beta$) and the carbon-climate ($\gamma$) feedbacks show a
considerable hysteresis behavior during negative emission phases. Hysteresis is stronger for the ocean
relative to the strength of the feedbacks, although the hysteresis of the terrestrial carbon cycle
feedbacks is larger in absolute terms. The well-known reduction of ocean and land carbon uptake with
increasing temperatures continues long into the negative emissions phases of the simulations (when
temperature is decreasing), albeit at a reduced rate. For the ocean, there is still a reduction in carbon
stocks due to legacy warming when pre-industrial atmospheric $CO_2$ is restored in the 1pctCO2
simulation, consistent with the single-model studies of Schwinger and Tjiputra (2018) and Bertini and
Tjiputra (2022). In contrast, all models agree that the effect of legacy warming is less important for the
terrestrial carbon-climate feedback as the reduction of global mean surface temperature leads to a
reduction in temperature-induced losses of terrestrial carbon towards the end of the 1pctCO2
simulation.
It is well known that carbon cycle feedback metrics vary over time, and between different scenarios.
Here we find that when (implied) emissions change from positive to negative, $\beta$ and $\gamma$ (defined
according to Friedlingstein et al. 2003) show an increase in absolute values due to the large hysteresis
of carbon stock changes, while temperature and atmospheric $CO_2$ decrease. Particularly, if the
deviations in surface temperature and atmospheric $CO_2$ become small towards the end of a modeled
negative emission scenario, the magnitude of these feedback metrics "explodes" since they are defined
as the ratio between the deviations in carbon stocks and the change in temperature and atmospheric
$CO_2$, respectively. Arguably, the latter is mainly a problem due to the strongly idealized simulation
design of the 1pctCO2 experiment, not for more realistic scenarios as the SSP5-3.4-OS. The feedback
metrics B and Γ (defined according to Boer and Arora, 2009), which are based on instantaneous fluxes,
also become ill-defined when deviations of surface temperature and atmospheric $CO_2$ approach zero,
but unlike the $\beta$ and $\gamma$ feedback metrics, they are only indirectly affected by the history of carbon fluxes.
These metrics thus respond faster to changes in atmospheric $CO_2$ concentration or temperature, for
example, B clearly shows the point in time when carbon fluxes reverse and the land or ocean turn from
a sink to a source of carbon under negative emissions.
We find that the relative strength of the feedback remains relatively robust between positive and
negative emission phases and between the different simulations considered here. For example, a model
with a stronger than average terrestrial carbon-concentration feedback ($\beta_L$) during the positive
emission phase of the 1pctCO2 simulation will also show a stronger than average $\beta_L$ during the negative
emission phase or for the SSP5-3.4-OS scenario. Regarding the model uncertainty of feedback metrics
we find that there is an increase in uncertainty in all feedback metrics between the positive and
negative emission phases of the 1pctCO2 simulation. Except for $\gamma_L$, this increase is much larger than
expected from an accumulation of uncertainty over time. This indicates that there is an additional
component of model uncertainty resulting from uncertainties in the model responses to the change
from increasing to decreasing radiative forcing.



The geographical patterns of terrestrial $\beta$ and $\gamma$ feedback metrics highlight differences in the responses
of tropical/subtropical versus temperate/boreal ecosystems as a major source of model disagreement.
For individual models, however, the spatial feedback patterns are remarkably similar during phases of
increasing $CO_2$ compared to phases of decreasing $CO_2$ concentrations, indicating that the increase of
global mean values of $\beta$ and $\gamma$ during negative emissions phases does not stem from a particular region
but is generally seen over the whole globe. We estimate the contribution of permafrost carbon release
to the carbon-climate feedback only for one of the five ESMs (NorESM2-LM, which vertically resolves
soil carbon). Permafrost carbon release is clearly seen as a strong negative feedback over the
permafrost area, but it emerges only relatively late in the simulations. Permafrost carbon release
accounts for 38% of NorEMS2-LM's carbon-climate feedback at the midpoint of the negative emission
phase of the 1pctCO2 simulation. NorESM2 has the lowest transient climate response of the ESMs
considered here and we therefore expect that other models might show an earlier and larger
permafrost carbon release.
In the SSP5-3.4-OS simulation, the presence of land-use change complicates the analysis of feedbacks.
Land-use change is not a feedback process, yet owing to the C4MIP simulation design, carbon losses (or
gains) due to land use change are confounded with the carbon-concentration feedback derived from a
biogeochemically coupled scenario simulation. If we disregard agricultural areas, terrestrial carbon
cycle feedback patterns in the SSP5-3.4-OS scenario are largely similar to those in the 1pctCO2
simulation.
We conclude with some recommendations for future research and the design of future model
intercomparison projects (MIPs) like C4MIP and CDRMIP. We expect that understanding and reducing
the large uncertainties in the response of ESMs to changes in atmospheric $CO_2$ and surface climate,
particularly during phases of negative emissions, remains a research topic of high relevance. Here, we
have shown that the uncertainties (model disagreement) in feedback metrics increases during phases
of negative emissions, and that this increase, for most of the feedback metrics, cannot be explained by
a linear accumulation of uncertainty with progressing simulation time. Identifying and better
understanding the causes of such increased model disagreement under negative emissions should be
pursued further with high priority.
Both the integrated-flux ($\beta$ and $\gamma$) and instantaneous-flux (Β and Γ) based feedback metrics were
designed at a time when nearly all future climate change scenarios were characterized by continuously
increasing atmospheric $CO_2$. Indeed both metrics perform well for such scenarios and have allowed us
to compare the strength of carbon-concentration and carbon-climate feedbacks across models, albeit
with their well-known caveats (e.g., their scenario dependence). However, in scenarios where
atmospheric $CO_2$ concentration decreases, these metrics become difficult to interpret, particularly in
the extreme case when atmospheric $CO_2$ concentration and surface temperature approach their pre-
industrial level. In the light of the discussion around CDR perhaps it is timely to rethink other but related
forms of these metrics that describe the response of land and ocean carbon systems in scenarios of
decreasing atmospheric $CO_2$ in a more robust manner.
The 1pctCO2 simulation combined with the 1pctCO2-cdr simulation is an extremely idealized model
experiment with huge (and infeasible) amounts of implied net-negative emissions and a discontinuity



at year 140, where implied emissions jump from large positive to large negative values. As we know
that carbon cycle feedbacks are scenario dependent, it would be preferable to assess these feedbacks
using model simulations that have a more realistic emission pathway and that include more realistic
amounts of net-negative emissions. Alternative idealized simulation designs that include negative
emissions have been proposed in the literature (MacDougall 2019; Schwinger et al. 2022) and we have
also considered the SSP5-3.4-OS scenario in this study. However, the presence of land-use change and
variable non-$CO_2$ forcings in SSP scenarios complicates the quantification of carbon cycle feedbacks.
Whether this problem can be solved for future phases of C4MIP by providing more detailed model
output or by requesting additional idealized experiments should be discussed in the C4MIP community.
Finally, most proposed negative emission options would be realized by manipulating the terrestrial or
oceanic carbon sinks (e.g., bioenergy with carbon capture and storage, afforestation or ocean
alkalinization), thereby not only changing the atmospheric $CO_2$ concentration and possibly the surface
climate but also the carbon cycle feedbacks themselves. Such interactions go beyond what can be
addressed with the traditional C4MIP design of fully- and biogeochemically coupled ESM simulations.
Consequently, a new framework for determining feedbacks in realistic scenarios of CDR deployment is
needed and should be developed in close collaboration with the integrated assessment modeling
community that will create such scenarios.
**Data availability**
All CMIP6 model output data is freely available through the Earth System Grid Federation (for example,
under https://esgf-data.dkrz.de/search/cmip6-dkrz/). The model output data of the 1pctCO2-cdr-bgc
simulation will be made publicly available upon final acceptance of this manuscript.
**Competing interests**
None of the authors has any competing interests.
**Acknowledgements**
A.A., J.S., and H.L. were supported by the Research council of Norway through the project IMPOSE
(grant no. 294930). J.S. and H.L. also received funding from the European Union's Horizon Europe
research and innovation programme (project RESCUE, grant agreement no. 101056939).
Supercomputing and storage resources for additional NorESM2 simulations were provided by UNINETT
Sigma2 (projects nn9708k/ns9708k). T.H. was supported by the Integrated Research Program for
Advancing Climate Models (TOUGOU, grant number JPMXD0717935715) and the Program for the
Advanced Studies of Climate Change Projection (SENTAN, grant number JPMXD0722681344) from the



Ministry of Education, Culture, Sports, Science and Technology (MEXT), Japan. C.D.J. and S.L. were
supported by the Joint UK BEIS/Defra Met Office Hadley Centre Climate Programme (GA01101), and
the European Union's Horizon 2020 research and innovation programme under Grant Agreement No
101003536 (ESM2025 - Earth System Models for the Future). R.S. and Y.S.-F. are grateful for the support
of the team in charge of the CNRM-CM climate model. Supercomputing time was provided by the
Meteo-France/DSI supercomputing center. R.S. acknowledges the European Union's Horizon 2020
research and innovation program under grant agreement No. 101003536 (ESM2025 – Earth System
Models for the Future). Y.S.-F. acknowledges the TRIATLAS project under the grant agreement No
817578 and the COMFORT project under the grant agreement No 820989.
We acknowledge the World Climate Research Programme, which, through its Working Group on
Coupled Modelling, coordinated and promoted CMIP6. We thank the climate modeling groups for
producing and making available their model output, the Earth System Grid Federation (ESGF) for
archiving the data and providing access, and the multiple funding agencies who support CMIP6 and
ESGF.
The work reflects only the authors' view; the European Commission and their executive agency are
not responsible for any use that may be made of the information the work contains.

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
