# Peer review of "Carbon cycle feedbacks in an idealized and a scenario simulation of negative emissions in CMIP6 Earth system models"

_EGUsphere, 2023_

## Author Comment (AC1)

**Authors' response to the reviewer #1**

We thank reviewer #1 for the positive assessment of our manuscript and for providing constructive and valuable criticism. We have carefully revised our manuscript, addressing each point raised as outlined in our point-by-point response below (original comments in gray, italic font). Proposed verbatim alterations or additions to the manuscript are highlighted in red.

*The authors explore how land and ocean carbon sinks react to rising and falling atmospheric CO2 levels and associated temperature changes in five Earth System Models. They examine both a stylized scenario and a more realistic overshoot scenario, finding a large hysteresis in the sink response and an ill-definement of the metrics when atmospheric CO2 approaches its initial level.*

***General comments***

*Overall, the paper is well-written and involves a large amount of work. I believe it is suitable for publication in Biogeosciences after some revisions.*

Thank you for the positive evaluation.

*Here are some suggestions to improve the manuscript:*

- *I am sceptical on how the authors tried to remove the effects from land-use change. Firstly, instead of removing gridcells with cropland fractions above a specific threshold (thereby removing gridcells with stable land cover over time), wouldn't it make more sense to remove gridcells with large changes in cropland fraction? Secondly, it seems the authors only considered croplands but not grazing land. Depending on how the ESMs implemented transitions between natural land and grazing land, this might cause substantial land-use emissions or carbon uptake (from forest regrowth on abandoned grazing land) in some regions. Thirdly, it would be good to compare your calculated metrics for the selected gridcells to all gridcells in the idealized simulations to see whether the selected gridcells are indeed representative for the entire globe.*

We agree that land-use changes (LUC) are a complication in the context of carbon cycle feedbacks, particularly since only limited LUC related ESM output has been made available for CMIP6. There have been two excellent studies published prior to ours (Melnikova et al. 2021, 2022) that discuss land use changes and carbon cycle feedbacks in the ssp534-over scenario in depth.

Our intention with including ssp534-over in our study was to compare the carbon cycle feedbacks on natural land only. To achieve this, we have to choose some threshold for crop-fraction below which we consider a gridcell as "dominated by natural land". Although the selection of the 25% threshold is arbitrary, it has been chosen following a series of sensitivity analyses. It serves as a representation of the "maximum" cropland fraction spanning from 2015 to 2100. To clarify, our methodology already includes gridcells with substantial shifts in cropland fraction. Conversely, those with changes resulting in a fraction consistently below 25% are regarded as minor changes (i.e., stable land cover over time). By taking *changes* in crop fraction over time as a criterion, we would have kept grid cells with large (> 25% in 1850) but stable crop fraction (which are discarded in our approach). However, the number of grid cells and associated surface area where this is the case is small, such that our results do not depend on this choice.

In response to your second point, we acknowledge that we do not treat pasture grid cells (or transitions from other land use types to pasture) explicitly in our approach, and that this was not discussed

appropriately in our manuscript. In ssp534-over, the strong expansion of bioenergy crops after 2040 is assumed to replace pasture (to avoid carbon emissions from deforestation). After the expansion of bioenergy crops into pasture land is completed around 2070, pasture area remains very stable in this scenario (see O'Neill et al. 2016, Fig. 4). Therefore, the majority of grid cells with transitions of pasture to cropland will be captured by our approach, while other transitions involving pasture are small in the ssp534-over scenario. We therefore believe that our approach that neglects treating pasture grid cells separately is defensible, but we will discuss this better in a revised version of our manuscript, added to the end of Section 2.2, as follows: "We acknowledge that our approach does not explicitly address pasture gridcells or transition from other land use types to pasture. Nonetheless, in the ssp534-over scenario, a substantial expansion of bioenergy crops between 2040 and 2070 is assumed to replace pasture areas, remaining relatively stable thereafter (see O'Neill et al. 2016). Hence, our approach captures the majority of such gridcells with transitions from pasture to cropland."

Regarding your third point, the metric (cumulative carbon uptake) used here, is of course sensitive to the total area. The selected gridcells therefore show a somewhat smaller carbon accumulation than the "entire globe". Nevertheless, regarding the overall shape and relative model differences the Figure AC1 below suggests that the selected gridcells indeed are representative of the entire globe (except for the CanESM5 model that appears to behave as an outlier in cropland areas as already mentioned in the text).

[Figure]

**Figure AC1:** Terrestrial carbon cycle feedbacks in the 1pctCO$_2$ and 1pctCO$_2$-cdr simulations for the entire globe (left column) and natural land (right column). An 11-year moving average has been used in all panels. Color coding is the same as Fig. 4 in the manuscript.

- *The authors find a large hysteresis in the sink response to declining atmospheric CO2 levels. This is likely a result of sinks responding not only to decreasing CO2 but are also still affected by the*

*previous CO2 increase (e.g. Chen et al., 2019; Chimuka et al., 2023; Krause et al., 2020). It would be good to discuss (and if feasible investigate) this more in the paper.*

We agree that the carbon cycle responses will be influenced by the effects of prior $CO_2$ increases. Also in response to comments by reviewer #2, we have expanded Section 3.2.3 by a discussion of the main causes for the hysteresis as follows: "For the ocean carbon cycle, hysteresis in the carbon-concentration feedback occurs mainly due to the long time scales of ocean overturning circulation. Schwinger and Tjiputra (2018) have shown that hysteresis strongly increases with water mass age. Young waters, which reside close to the ocean surface, exchange quickly with the atmosphere and show little hysteresis, whereas old, deep ocean water masses' responses to declining atmospheric $CO_2$ will be delayed, and thus show considerable hysteresis. Over land, both the vegetation and soil carbon pools show a lagged response to decreasing $CO_2$ due to the fact that transient changes in $[CO_2]$ lead to a long term disequilibrium between the $CO_2$ fertilization effect, vegetation biomass, litterfall, and soil carbon (e.g., Krause et al. 2020). Therefore, despite declining $[CO_2]$ levels at the beginning of the ramp-down phase there is still an increase in vegetation biomass due to $CO_2$ fertilization, and consequently an increase in soil carbon due to still increasing litterfall. Warming-induced hysteresis appears to be larger for soil carbon in most models. Similar to the large warming induced hysteresis in the ocean, this is caused by the fact that even though warming levels start to decline shortly after the onset of the ramp-down phase, environmental conditions remain warmer than in the pre-industrial period over the whole time of the ramp-down simulation."

**Specific comments**

*L19: I wonder whether the 1.5° target is still realistically feasible, I suggest to change "1.5" to "well below 2°"*

This has been done.

*L31: Does this mean the growth is the same everywhere or that there are spatial variations but all models show the same patterns?*

It is intended to convey that although there are spatial variations, all models still exhibit the same general pattern. We reworded the text as "We find that this growth over time occurs such that the spatial patterns of feedbacks do not change significantly for individual models."

*L107: I thought the Arctic is warming at least 3x faster, please double-check. Also the references seem to be model-based.*

We have added other references and reworded the text as "... that Arctic temperatures are increasing at a much faster rate than the global average (Liang et al. 2022; Rantanen et al. 2022) …"

*L118ff: Include Chimuka et al. (2023). Also it's not entirely clear to me how the study goes beyond those previous studies (e.g. SSP535 is also used in Melnikova et al.). Can you elaborate a bit more?*

We have added a reference to Chimuka et al. (2023). We have also included a discussion of the work by Chimuka et al. (2023) in several places in the revised manuscript following other comments.

Our study is different from Melnikova et al. (2021, 2022) because we focus on the $1pctCO_2$ and $1pctCO_2$-cdr simulations. A biogeochemically coupled $1pctCO_2$-cdr simulation was not part of CMIP6 and

has been performed by 5 ESMs for this study. Therefore, we are able to provide an analysis of $CO_2$ and temperature induced feedbacks during the ramp-down phase of the 1pctCO$_2$-cdr simulation for the first time. We consider the ssp534-over scenario only to compare feedbacks on the fraction that is not affected by land-use changes with the 1pctCO$_2$ scenario.

*L169: "gases"*

Corrected.

*L239: Maximum for the average over the whole time period or maximum of individual years?*

The original crop fraction data for individual models is available in monthly timesteps. The maximum is calculated over the entire time series. In other words, we determine the highest value from the original crop fraction data for individual months over the period 2015-2100. To make this clearer we rewrite this sentence as follows: "... we define natural land as grid cells with a maximum cropland fraction of less than 25% at all time steps during the period 2015-2100."

*L239: "cropland"*

Corrected.

*L240: remove comma*

Done!

*L254: This seems a major limitation, I assume models without dynamic vegetation to have a lower hysteresis. In general, in addition to calculating the different terms (Fig. 7), can you say more on which differences in modelled processes you think drive the differences in the carbon response to decreasing CO2?*

We agree that the inclusion of dynamic vegetation should lead to a larger hysteresis of land carbon pools during an overshoot. However, results from our small model ensemble remain inconclusive, since it seems that model uncertainty stemming from a variety of processes is larger than this effect. This is highlighted in Fig. 4 which demonstrates that although UKESM1-0-LL has a strong hysteresis of the carbon-climate feedback, CanESM5 (without dynamic vegetation) has an even larger carbon-climate feedback.

Also regarding other differences in modelled processes, the results of our calculations presented in Fig. 7 are less conclusive than we had hoped for. For example, the three models that include a representation of the nitrogen cycle ("nitrogen limitation", MIROC-ES2L, NorESM2-LM, UKESM1-0-LL) do not show generally a lower $CO_2$ fertilization effect.

*L256: Remove comma.*

Done!

*L268: Another comma I think should be removed.*

Done!

*Fig. 1: Please use consistent names (not only here but throughout the manuscript – there is a mixture e.g. of "SSP5-3.4-OS", "ssp534-os" and "ssp534-over"), avoid too many abbreviations ("ROC"), use subscripts for CO2. "1pctCO2" is instead a combination of "1pctCO2" and "1pctCO2-cdr", I don't think this has been made clear. Maybe term this combination "1pctCO2-1pctCO2cdr" (or simply "1pctCO2-cdr" as it is an extension of "1pctCO2") and use this name throughout the manuscript?*

To ensure clarity and consistency, we have now used the naming "ssp543-over" throughout the manuscript, used subscripts for "CO2", and removed the "ROC" abbreviation from Fig. 1. Furthermore, we clarified the namings associated with the $1pctCO_2$ experiment by adding "Unless otherwise noted, for the sake of simplicity, we will use "$1pctCO_2$-cdr" to refer to both positive and negative phases of this experiment ($1pctCO_2$ plus $1pctCO_2$-cdr) to the text on lines 236-238 of the revised manuscript.

*L326: "(Fig. 2c,d)" should be moved to end of the sentence.*

Will do, thank you!

*L329ff: Not sure I understand this interpretation, it sounds as if the carbon does not need to be released without climate warming. I think what you mean instead is that more initial carbon uptake happens in the BGC simulation and this extra carbon is then released? Also I think warming reduces the land carbon sink in SSP5-3.4-OS rather than increasing the source?*

Sorry, these sentences were not very clear. We reformulated them as "We therefore interpret the increased source of carbon at the end of the $1pctCO_2$-BGC simulation as a release of additional carbon that has been taken up in the absence of climate warming during the biogeochemically coupled simulation. The net negative emission phase of the ssp534-over scenario is too short to show this effect in 2100 (where the warming effect still reduces  the model mean terrestrial carbon sink)."

*L332: Remove second comma.*

Done!

*L414ff: I find this unclear. Is the mechanism that "cropland grid cells" in this scenario tend to keep/increase their cropland fraction until the end of the century so they don't benefit much from CO2 fertilization?*

Yes, this is correct. We make this point clearer by rewording this sentence as follows: "This indicates, consistent with Melnikova et al. (2022) who demonstrate that carbon losses from land use changes dominate over gains through $CO_2$ fertilization in crop dominated areas (see their Fig. 4, panels a and c), that the prescribed …"

*L416: I understand what you mean but one could argue land-use change is affected by atmospheric CO2. Change e.g. to "We note that land use change is an input rather than a feedback process in our simulations."*

We reworded this sentence as suggested: "We note that land use change is externally prescribed rather than a feedback process in our simulations."

*L431: I am not sure I understood how the numbers were calculated. Is it the cumulative carbon flux between year 70 and 210 (so without hysteresis and perfect reversibility it would be 0)?*

Yes, this is correct, our definition of hysteresis is the difference between the two points on the ramp-up and ramp-down pathways in individual models intersected by a vertical line at 570 ppm. We see that we were not very clear on the definition of hysteresis, and we added the following text: "...or returned to this value after the overshoot. We define hysteresis as the difference between cumulative carbon uptake in year 210 minus cumulative carbon uptake in year 70 (i.e., hysteresis is positive, if cumulative carbon uptake is *larger* on the ramp-down side of the 1pctCO2 simulation)."

*Fig. 5,8: Are b) and e) really needed?*

We believe that panels b and e with smaller scales on the y-axis are useful since they facilitate the comparison of results between the SSP-scenario and the 1pctCO$_2$ simulation as well as with previous studies that only calculated feedback factors for the ramp-up phase.

*Fig. 7c: ".ppm" should be "/ppm". Can you also add a legend?*

The dot is removed. A legend has been added to the figure.

*Fig. 8: Change "fdbk" to "feedback".*

"fdbk" has been updated to "feedback", thank you!

*L628: But other carbon losses would likely also increase for a higher TCR, so the contribution might be the same?*

Thank you, this is a good point. Other models involved in this study, which are characterized by higher TCR (e.g., UKESM1-0-LL and CanESM5) cannot resolve permafrost carbon, so unfortunately we cannot compare their permafrost carbon losses versus other carbon losses due to warming. It might well be that the percentage contribution of permafrost losses to the total feedback remains similar also for higher TCRE models, but this is highly speculative. Since we are not able to provide any evidence for or against this hypothesis, we would prefer to leave this sentence as it is.

*L629: Typo NortESM2-LM.*

Thank you, corrected.

*Fig. 9: The white areas (croplands) in the right figure are hardly distinguishable from areas with small changes. Consider using grey colour instead.*

Thank you for your suggestion. We have used gray color for the (masked out) cropland areas to make them distinguishable from areas with small changes in the figure.

*L686: These grid cells seem pretty far north to be used for croplands. Can you checker whether there is indeed some cropland expansion happening there? Could it also be related to wood harvest which may only be represented in UKESM and CanESM?*

Thank you, we have double checked this and these latitudes are indeed too far north for cropland. Wood harvest is only represented in NorEMS2 and MIROC-ES2L (not in CanESM5 and UKESM1-0-LL). We have investigated the high latitude negative $\beta$-values further and attributed this effect to the warming pattern caused by non-CO$_2$ forcing. We have rewritten the text following line 682 as follows: "These differences are related to the negative $\beta$-values (discussed above) for these models, which make the carbon gain due to warming (the difference $\Delta C^{cou} - \Delta C^{bgc}$) considerably larger than in the 1pctCO$_2$ simulation. Again, this is reinforced by the fact that the global average temperature change in the ssp534-over simulation is positive and thus ($\Delta T^{cou} - \Delta T^{bgc}$) is smaller than the actual (local) temperature differences. This indicates that, if the global mean temperature change due to non-CO$_2$ forcings does not broadly reflect local changes correctly (e.g., local cooling vs. global warming), regional scale feedback factors might show unexpected results."

*L733: "in the SSP5-3.4-OS scenario". Do you have an idea why the uptake does not occur in 1pctCO2?*

This difference in high latitude $\gamma$-values in NorESM2-LM and UKESM1-0-LL is directly related to the negative $\beta$-values in the same region. Since there is carbon loss in the ssp534-over simulation in these models, the carbon gain due to warming (the difference $\Delta C^{cou} - \Delta C^{bgc}$) becomes larger than in the 1pctCO$_2$ simulation.

*L743: This section is very long for a summary and conclusion section. Consider shortening.*

Thank you for your feedback. We will work on shortening this section.

*L799: Is this really a negative feedback or rather a lag in a positive feedback?*

Thank you for bringing this mistake to our attention. We have corrected the text as follows: "strong positive feedback (i.e., negative γ)".

*L840: Do you have suggestions for such experiments?*

For example, to disentangle the effects of land use change, requesting a biogeochemically coupled simulation with fixed land use could be useful. Such simulations (not biogeochemically coupled though) have been done for CMIP6 LUMIP (Land Use Model Intercomparison Project). We added this suggestion to this sentence "(e.g., scenario simulations with fixed land use)".

*L844f: I don't understand, isn't it these interactions that make SSP5-3.4 so difficult to interpret? Why would it not be possible to include e.g. afforestation?*

Negative emission options such as afforestation or ocean alkalinization will increase the terrestrial or oceanic carbon sink. At the same time, the additional carbon uptake will have a feedback on the whole carbon cycle. These feedbacks cannot be determined with the "traditional" C4MIP set-up. We make this clearer by extending the sentence: "Consequently, a new framework for determining feedbacks caused by large scale CDR in realistic scenarios of CDR deployment is needed and should be developed…"

References

Chimuka, V. R., Nzotungicimpaye, C.-M., and Zickfeld, K.: Quantifying land carbon cycle feedbacks under negative CO2 emissions, Biogeosciences, 20, 2283–2299, https://doi.org/10.5194/bg-20-2283-2023, 2023.

Krause, A., Arneth, A., Anthoni, P., and Rammig, A.: Legacy Effects from Historical Environmental Changes Dominate Future Terrestrial Carbon Uptake, Earth's Future, 8, e2020EF001674, https://doi.org/10.1029/2020EF001674.

Liang, Y.-C., L. M. Polvani, and I. Mitevski, 2022: Arctic amplification, and its seasonal migration, over a wide range of abrupt CO2 forcing. Npj Clim. Atmospheric Sci., 5, 14, https://doi.org/10.1038/s41612-022-00228-8.

Melnikova, I., and Coauthors, 2021: Carbon Cycle Response to Temperature Overshoot Beyond 2°C: An Analysis of CMIP6 Models. *Earths Future*, **9**, e2020EF001967, https://doi.org/10.1029/2020EF001967.

Melnikova, I., and Coauthors, 2022: Impact of bioenergy crop expansion on climate–carbon cycle feedbacks in overshoot scenarios. *Earth Syst. Dyn.*, **13**, 779–794, https://doi.org/10.5194/esd-13-779-2022.

O'Neill, B. C., and Coauthors, 2016: The Scenario Model Intercomparison Project (ScenarioMIP) for CMIP6. Geosci. Model Dev., 9, 3461–3482, https://doi.org/10.5194/gmd-9-3461-2016.

Rantanen, M., Karpechko, A.Y., Lipponen, A. *et al.* The Arctic has warmed nearly four times faster than the globe since 1979. *Commun Earth Environ* 3, 168 (2022). https://doi.org/10.1038/s43247-022-00498-3.

Schwinger, J., and J. Tjiputra, 2018: Ocean Carbon Cycle Feedbacks Under Negative Emissions. Geophys. Res. Lett., 45, 5062–5070, https://doi.org/10.1029/2018GL077790.

---

## Author Comment (AC2)

**Authors' response to the review by Kirsten Zickfeld**

We thank Kirsten Zickfeld for the positive assessment of our manuscript and for providing constructive and valuable criticism. We have carefully revised our manuscript, addressing each point raised as outlined in our point-by-point response below (original comments in gray, italic font). Proposed verbatim alterations or additions to the manuscript are highlighted in red.

*Asaadi et al. quantify carbon cycle feedback in scenarios with positive and negative emissions using an ensemble of ESMs. They use cumulative and flux-based carbon cycle feedback metrics to quantify global as well as regional carbon cycle feedbacks. They also use a decomposition approach to quantify the contribution of various carbon cycle processes to carbon-concentration feedback strength. They find that the carbon-concentration and carbon-climate feedbacks as well as the uncertainty in these feedbacks increase during the negative emissions phase.*

*The manuscript is a valuable contribution to a small existing body literature on carbon-cycle feedbacks under negative CO2 emissions. It is mostly well written, clearly structured and informative. There are, however, several aspects that need to be addressed before the paper is acceptable for publication in Biogeociences.*

Thank you for this positive evaluation.

- *Non-CO2 radiative forcing in SSP3.4-OS simulations. The inclusion of non-CO2 radiative forcing in SSP5-3.4OS simulations hampers the comparability of carbon cycle feedbacks with the idealized 1pctco2 simulation. The temperature changes induced by non-CO2 forcings are significant in some models and the response to these changes in the BGC simulations confounds the response to changes in atmospheric CO2. Given this complication, along with the effect of residual land-use changes and the small period of net negative emissions in SSP5-3.4OS, I would like to encourage the authors to think about whether inclusion of this scenario in the paper is warranted. The paper is quite long and a stronger focus would be beneficial. If the authors decide to retain analysis of the SSP3.4-OS simulations in the manuscript, a more in-depth discussion of the non-CO2 induced temperature effects is needed.*

We agree that the complications related to non-$CO_2$ radiative forcings in the ssp534-over scenario were not discussed in enough detail. The feedback framework as used here is in principle suited to be applied to simulations with non-$CO_2$ forcings as long as the assumption of linearity holds (see discussion in proposed new text). We believe that adding a concise discussion of this issue (rather than removing the ssp534-over scenario altogether) would strengthen our manuscript. We propose to add a short paragraph after line 178 of the preprint manuscript as follows:

"By combining equations (1) and (2) to yield

$$\beta_X = \frac{1}{\Delta[CO_2]}\left(\Delta C_X^{BGC} - \gamma_X \Delta T^{BGC}\right) \qquad (3)$$

it can be seen that, in order to calculate $\beta_X$, the carbon stock changes in the biogeochemically coupled simulation are corrected for global mean temperature changes using $\gamma_X$. Hence, temperature changes in the biogechemically coupled simulation are fully accounted for as long as the underlying assumption of linearity holds. However, this assumption might be problematic, for example, if the spatial pattern of warming in a biogeochemically coupled scenario simulation arising from non-$CO_2$ forcings is very

different from the warming patterns in the fully coupled simulation, particularly if the sign of the local temperature change is different from the global average (e.g., local cooling vs. global average warming). Such effects could become important on regional to local scales and will be discussed in Section 3.4."

We do see effects of non-$CO_2$ forcing in the regional $\beta$- and $\gamma$-values (Figs. 9 and 10), most notably negative $\beta$-values in high latitudes in some models that are not found in the 1pct$CO_2$ simulations (NorESM2-LM and UKESM1-0-LL, and to a lesser extent CanESM5). We attribute this difference to the very different spatial pattern of temperature changes in some models in the ssp534-over compared to the 1pct$CO_2$ simulation (see Fig. S9 below, which will be added to the supplementary; note the figure shows normalized temperature changes $\Delta T / \overline{\Delta T}$). In NorESM2-LM, UKESM1-0-LL, and CNRM-ESM2-1, the ssp534-over BGC simulation shows local cooling, which is only marginally present in the fully coupled simulation. This (together with other changes in local climate) can lead to local carbon losses and negative $\beta$-values. In NorESM2-LM and UKESM1-0-LL, these negative values are then reinforced by positive $\gamma$-values in this region and positive global mean temperature change via equation 3. This indicates that, in the case of non-$CO_2$ forcings (particularly aerosol forcing, which is regionally fragmented) the global mean temperature change is not a good proxy for regional climate changes.

We will revise and expand the discussion on the effects of non-$CO_2$ forcing in Section 3.4 (from line 682 of the preprint and after line 732). Please see our detailed response to the corresponding specific comment below.

[Figure]

**Figure S9**: ΔT normalized by its global mean value for individual models. Temperature deviations are averaged over 21-year time intervals centered on the year 70 for the fully coupled 1pctCO$_2$ experiment and the year 2045 both for the fully and biogeochemically coupled versions of the ssp534-over scenario. The BGC version of the ssp534-over simulation represents non-CO$_2$ induced radiative forcing, along with the effects of land-use changes. The fully coupled 1pctCO$_2$ represents only CO$_2$ induced warming.

- *Hysteresis: The manuscript describes in the hysteresis in the integrated land and ocean carbon flux response to changes in CO2 and temperature, but misses to provide an explanation for why this hysteresis may occur. Identification of possible causes (e.g. lagged response to forcing) could help to explain some effects described in the paper (see specific comments).*

Thank you for this suggestion, we have revised our manuscript accordingly and discuss possible causes of hysteresis in more detail. Please see our detailed responses to the specific comments below.

- *Calculation of feedback metrics: The authors chose to calculate the feedback metrics during the negative emissions phase using the same reference year (pre-industrial) as for the positive emissions phase. This leads to feedback metrics being ill-defined as the pre-industrial state is approached in the ramp-down phase of the 1pctco2 simulation. An alternative approach has been proposed (Chimuka et al., 2023) that uses the time of transition from positive to net negative emissions as the reference year. The advantage of such an approach is that it quantifies carbon cycle feedbacks specifically under conditions of declining atmospheric and cooling, which is consistent with the stated objective of the manuscript (l. 134-145). These alternative approaches should at least be acknowledged and discussed.*

We agree that this interesting new approach has to be acknowledged and discussed in our paper. We have introduced a new paragraph in the introduction (after line 101 of the preprint manuscript), which reads as follows: "One open question regarding carbon cycle feedbacks under negative emissions is relative to which state the feedbacks should be measured. A sensible definition requires that any gain or loss of carbon is calculated relative to a state where the carbon cycle is in equilibrium. Schwinger and Tjiputra (2018) have opted to keep the pre-industrial state as the reference also after the onset of negative emissions. We follow this approach here, but we note that recently Chimuka et al. (2023) proposed an alternative approach, which defines the feedbacks during the negative emission phase relative to the state at the onset of negative emissions. Since, the Earth system will be in disequilibrium at this point in time, this approach requires additional simulations (e.g., 500 years of zero-emission simulation initialized from the peak of [CO2] at the beginning of the ramp-down phase) that allow to estimate and remove the lagged response of the Earth system to this disequilibrium."

- *Description of model results lacks precision in some instances (see specific comments).*

Thank you for your feedback. We improve the description of our results, addressing the specific comments you provided.

**Specific comments**

*l 19: The goal of the Paris Agreement is to limit warming to "well below 2°C" above pre-industrial levels.*

We have replaced "Limiting global warming to 1.5°C…" by "Limiting global warming to well below 2°C…"

*l 45, "exhausted within the next few decades". Decades -> years. Include reference to updated carbon budget estimates in Forster et al., 2023.*

We have revised "decades" to "years" and included a reference to the updated carbon budget estimates in Forster et al. (2023).

*l 50-51: Include more recent references, e.g. IPCC Synthesis Report, State of CDR report (Smith et al., 2023).*

More recent references are included, including the IPCC Synthesis Report and the State of CDR report (Smith et al., 2023) and Forster et al. (2023).

*l 118: A recently published study by Chimuka et al. explores land carbon cycle feedbacks under negative emissions. Please include reference.*

The reference has been included now. Thank you! See also our response above.

*l 138-139: "We briefly explore … the impact of alternative metrics". The need for alternative metrics in mentioned in the Conclusions, but the paper does not include an exploration of these metrics. Please rephrase.*

Here we referred to the instantaneous flux-based approach described in Section 2.1, as presented by Boer and Arora (2009). The text has been revised as follows: "We also briefly …the impact of alternative feedback metric definitions that rely on instantaneous carbon fluxes rather than carbon stocks in the context of negative emissions ."

*l 173-174: Please clarify whether the assumption DT_BGC=0 was used in the calculation of feedback metrics for 1pctCO2 and SSP5-3.4-OS simulations; for the latter this assumption is not justified due to non-CO2 forcings applied in the BGC simulation.*

We did not use the assumption DT_BGC=0. All feedback factors (global and regional) were calculated using the full expression for gamma and beta. Only for comparison and to assess what impact this assumption would have, results based on the assumption of DT_BGC=0 are shown as dotted curves in Fig. 5d and Fig. 8. The impact of this assumption was relatively small on global average. We clarify this by adding the sentence: "All results presented here are calculated using the complete expression for $\beta$ and $\gamma$ (without the assumption $\Delta T^{BGC} = 0$). For comparison, we also provide feedback factors calculated using the simplified (rightmost) definition of $\beta$ and $\gamma$ in some figures."

*l 306: "smaller magnitude of the temperature anomaly". I think this should read "larger magnitude".*

Corrected, thank you!

*l 308-309; "suggests that a substantial part of the carbon-climate feedback…". Unclear how you reach this conclusion. Please explain.*

This is based on the magnitude of temperature changes in the BGC simulation: For 4 out of five models, the peak warming in the BGC simulation reaches about 10-30% of the peak warming in the fully coupled simulation. Assuming that land use change is not a strong driver of global average (non-local) temperature changes, we conclude that the non-$CO_2$ forcings contribute "substantially" to the temperature changes, and thus will cause a substantial part of the carbon-climate feedback. We agree that we could make our statement more quantitative: "The relatively large magnitude of the temperature anomaly in the BGC simulation of the ssp534-over scenario (peak warming of 12% - 29% of the peak warming in the fully coupled simulation) suggests that warming due to non-$CO_2$ forcings might contribute substantially to the carbon-climate feedback in the ssp534-over scenario."

*l 324-325: "the terrestrial CO2 source … is larger….": Models with a larger terrestrial sink have a larger source in the ramp-down phase of the BGC simulation. This suggests that these models have a larger sensitivity (DC_L/DCO2) to both atmospheric CO2 increase and decrease.*

Thank you, we have added this as follows: "We also observe (Fig. 2c,d) that models which take up more (less) terrestrial carbon during the $CO_2$ ramp-up phase (1pct$CO_2$) release more (less) carbon towards the end of the $CO_2$ ramp-down phase (1pct$CO_2$-cdr-bgc), indicating that these models have a larger (smaller) sensitivity ($\Delta C_L/\Delta CO_2$) to both atmospheric $CO_2$ increase and decrease."

*l 369: Which "simulations"?*

We rephrased the text as "At the end of the ssp534-over and 1pct$CO_2$ simulations,... "

*l 372 (and elsewhere in sections 3.2.1 and 3.2.): "the ocean carbon-concentration feedback is larger…". Need to explain how the magnitude of feedbacks is inferred. I assume you are using the slope but his needs to be clarified.*

Thank you, we have clarified this by adding: "Generally, the ocean carbon-concentration feedback (as indicated by the cumulative carbon uptake per unit increase of $CO_2$ concentration, Fig. 3a-c) is larger in the ssp534-over scenario, which can most likely be explained with the slower growth rate of [$CO_2$] in this scenario compared to the 1pct$CO_2$ simulation." We add similar text in line 409 (section 3.2.2).

*l 408: It is worth pointing out that, in contrast to the ocean, the integrated atmosphere-land flux starts to increase, albeit with a lag, in response to cooling in the negative emissions phase in most models.*

Thank you for your suggestion. We have incorporated your input into the text as follows: "It is worth mentioning that, unlike the ocean, the COU-BGC accumulated atmosphere-land flux starts to increase, albeit with a lag, in response to cooling during the negative emissions phase in most models (Figs. 3e and 4e)."

*l 411-413: "This is because…". This needs to be explained and justified more clearly. Are you saying that because "cropland grid cells" have a smaller cumulative flux in the SSP-3.4-OS simulations, this can also be expected for grid cells with a cropland fraction <25%?*

Yes, this is what we intended to say. Within our "grid cells dominated by natural land", we can still have up to 25% crop fraction, and this fraction of the grid cells can be expected to behave similar to crop dominated grid cells. We have revised the text as follows: "...is the driver behind the small (negative for NorESM2-LM and UKESM1-0-LL) carbon accumulation for crop land grid cells. Since grid cells that are dominated by natural land according to our separation approach, may contain up to 25% croplands, we expect a reduction of cumulative carbon fluxes due the remaining land use (changes) in the natural land grid cells."

*l 415: "driver": How about the role of non-CO2 forcings in SSP5-3.4-OS?*

We see a very pronounced difference between grid cells with >=25% crop fraction and <25% crop fraction. When viewed spatially, $\beta$ becomes strongly negative in agricultural areas (not shown, we have masked "crop dominated" grid cells out in Fig. 9). Although we cannot rule out some influence of non-$CO_2$ forcers (particularly aerosols, see also our response above), we believe that in this context the mentioning of non-$CO_2$ forcings would be too speculative.

*l 424: "remains very similar": Several models show significant differences (MIROC, CanESM, UKESM).*

We apologize, the text was somewhat imprecise. We reworded the sentence as "... the model-mean carbon-climate feedback for cropland and natural land remains very similar between the ssp534-over and 1pctCO$_2$ simulations (Fig. S3f)."

*l 439-442: It would be helpful if the authors could point to potential causes for the hysteresis, such as lagged response to forcing and/or tipping points/state changes. This could also help with the interpretation of results. E.g. if the larger concentration-carbon feedback in some models is dominated by tree-PFTs (which appears to be the case based on the statement in l. 658-659), the longer response timescale of these PFTs could explain why models with a larger carbon-concentration feedback also have larger hysteresis.*

We have added a paragraph after line 448 of the preprint, summarizing the main causes for hysteresis as follows: "For the ocean carbon cycle, hysteresis in the carbon-concentration feedback occurs mainly due to the long time scales of ocean overturning circulation. Schwinger and Tjiputra (2018) have shown that hysteresis strongly increases with water mass age. Young waters, which reside close to the ocean surface, exchange quickly with the atmosphere and show little hysteresis, whereas old, deep ocean water masses can only respond to the declining atmospheric CO$_2$ when they are re-ventilated to the surface layer, which can take hundreds to thousands of years, and thus show considerable hysteresis. Over land, both the vegetation and soil carbon pools show a lagged response to decreasing CO$_2$ due to the fact that transient changes in [CO$_2$] lead to a long term disequilibrium between the CO$_2$ fertilization effect, vegetation biomass, litterfall, and soil carbon (e.g., Krause et al. 2020). Therefore, despite declining [CO$_2$] levels at the beginning of the ramp-down phase there is still an increase in vegetation biomass due to CO$_2$ fertilization, and consequently an increase in soil carbon due to still increasing litterfall. Warming-induced hysteresis appears to be larger for soil carbon in most models. Similar to the large warming induced hysteresis in the ocean, this is caused by the fact that even though warming levels start to decline shortly after the onset of the ramp-down phase, environmental conditions are warmer than in the pre-industrial period over the whole time of the ramp-down simulation."

[Figure]

**Figure AC1**: same as Fig. 4 in the preprint manuscript but for vegetation carbon cycle feedbacks.

[Figure]

**Figure AC2**: same as Fig. 4 in the preprint manuscript but for soil carbon cycle feedbacks.

*l 509-510: "This implies …": The fact that feedback metrics as calculated in this study become ill-defined at the end of the 1pctCO2 simulation is not a problem of the metrics themselves, but the choice of*

*reference year used for the calculation of the anomalies in the ramp-down phase. An alternative that addresses this problem is to use the transition year from positive to negative emissions as reference year (see Chimuka et al., 2023).*

Thank you for pointing this out, we mention this as follows: "We note that this problem is connected to the choice of the reference relative to which the feedbacks are calculated. In the approach of Chimuka et al. (2023), where the reference is chosen to be at the transition from positive to negative emissions, singularities towards the end of the 1pctCO$_2$-cdr simulation are avoided."

*l 516-517: To interpret this increase in model uncertainty it would be valuable if the authors could address the additional processes that become relevant in ramp-down phase. E.g. the uncertainties would be expected to increase if models exhibits different lag times in response to the prior CO2 increase.*

A bit further down, lines 573-576 of the preprint text, we already discuss reasons for the increased uncertainty in $\beta_L$: "It is worth noting that for four out of six terms of Eq. 3 ... the model disagreement is significantly larger during the ramp-down phase of the 1pctCO$_2$ simulation, indicating that changes in these processes are responsible for the strong increase in model uncertainty in $\beta_L$ between positive and negative emission phases pointed out in the previous section". We believe it would be confusing to add additional explanations around line 516 and would prefer to leave the text as it is.

*l 654.: "consistent with the lagged response": this is the first time a lagged response is mentioned. The possibility of such responses should be discussed earlier in the context of hysteresis.*

We have revised our manuscript to mention the "lagged response" earlier in Section 3.2.3 on hysteresis, please see our response above.

*l 683-685: This inference is incorrect. Calculating the metrics assuming DT_BGC=0 changes the value of the feedback parameter, but does not remove the confounding effect of non-CO2 induced warming on beta.*

We see that this statement was not well explained. We agree, there is an effect of non-CO$_2$ forcings on the carbon uptake $\Delta C_X^{BGC}$, regardless of how we calculate the feedback metrics. However, $\beta_X$ is corrected for the global average warming that might occur in the BGC simulation by using $\gamma_X$:

$\beta_X = \frac{1}{\Delta[CO_2]}\left(\Delta C_X^{BGC} - \gamma_X \Delta T^{BGC}\right)$ (Equation 3 in the revised manuscript, see also our response

above). Therefore, negative $\beta_X$ values could occur even if $\Delta C_X^{BGC}$ was positive. We calculated $\beta_X$ with the assumption $\Delta T^{BGC} = 0$ to check whether the carbon uptake is actually negative (which is the case).

We have revised and expanded the discussion of the negative $\beta_X$ values (and differences in $\gamma_X$, which are related) in the ssp534-over scenario and their relation to non-CO$_2$ forcings: "Unlike in the 1pctCO$_2$ experiment, temperature changes are not negligible in the BGC simulation of the ssp534-over experiment (Fig. 1). Furthermore, the spatial pattern of temperature changes is very different for some models, particularly for UKESM1-0-LL, NorESM2-LM, and CNRM-ESM2-1, which show local cooling that is not present (or much weaker) in the fully coupled simulations (Fig. S9). This cooling (and other changes in surface climate related to non-CO$_2$ forcings) lead to local carbon losses and negative $\beta$-values in

UKESM1-0-LL and NorESM2-LM in northern high latitudes. In addition, according to Eq. 3, these negative values are reinforced by positive $\gamma$-values in this region and a positive global mean temperature change in ssp534-over in these models (see Eq. 3). In contrast, CNRM-ESM2-1 does not show negative values of $\beta$ in northern high latitudes (despite local cooling), which can be explained by much larger $\beta$-values to begin with, and a smaller (and negative) temperature sensitivity $\gamma$ in high latitudes."

Further down (line 734) we have added: "These differences are related to the negative $\beta$-values (discussed above) for these models, which make the carbon gain due to warming (the difference $\Delta C^{cou} - \Delta C^{bgc}$) considerably larger than in the 1pctCO$_2$ simulation. Again, this is reinforced by the fact that the global average temperature change in the ssp534-over simulation is positive and thus ($\Delta T^{cou} - \Delta T^{bgc}$) is smaller than the actual (local) temperature differences. This indicates that, if the global mean temperature change due to non-CO$_2$ forcings does not broadly reflect local changes correctly (e.g., local cooling vs. global warming), regional scale feedback factors might show unexpected results."

*l 685-687: It would be helpful if cropland grid cells that were omitted in the global feedback metric calculations could be clearly identified in the maps in Figs. 9 and 10 for both the 1pctco2 and SSp5-3.4-OS simulations (e.g. by a contour line delineating these grid cells).*

Thank you for your suggestion. We have used gray color for the (masked out) cropland areas to make them distinguishable from other areas with small changes in the figure.

*l 700-701: "predominantly negative value of gamma_o": by closely looking at the maps it looks like gamma_o exhibits a banded pattern of positive and negative values.*

Thank you, we reworded our sentence as follows: " Figure 10 indicates that the ESMs considered here simulate predominantly negative values of $\gamma_o$ over the ocean. Positive values of $\gamma_o$ are found in the Arctic, and in the Southern Ocean most models simulate a banded pattern of positive (adjacent to Antarctica), negative (centered between 60 and 50°S), and positive (between approximately 50 and 40°S) values."

*l 756-757: "Hysteresis is stronger relative..". Sentence unclear. Please rephrase.*

We reworded this sentences as "At the same level of atmospheric CO$_2$ concentration, the ocean exhibits a larger hysteresis than its corresponding feedback during the ramp-up phase, while the terrestrial carbon uptake displays a larger hysteresis in absolute magnitudes."

*l 774-775: The singularity of beta and gamma at the end of the 1pctco2 simulation is not a problem of the experimental design but the choice of reference year.*

We have mentioned the alternative approach of Chimuka et al. in the revised version of our summary and conclusions.

*l 782-783: Unclear what is meant by "relative strength of the feedback".*

We have deleted this sentence in the revised version of the summary and conclusions, which was shortened in response to a comment of reviewer #1.

*l 789-790: This "additional component of uncertainty" could be the different response timescales exhibited by the models in response to prior forcing. See earlier comment.*

Thank you for your suggestion. We have incorporated the following statement into the text: "This additional component of model uncertainty can be attributed to the varying response timescales of individual models in response to the preceding forcing."

*l 799: "strong negative feedback": Unclear which feedback you are referring to.*

Thank you for bringing this to our attention. We concur that selecting precise wording is crucial; although the feedback itself is positive, the gamma values are indeed negative. We have adjusted the text as follows: "strong positive feedback (i.e., negative γ) ".

*l 805: Given these complications as well as the complications arising due to inclusion of non-CO2 forcing, what is the value of including these simulations in the feedback analysis?*

The value is mainly in describing these complications. Now that the next phase of C4MIP is being discussed, we believe it is valuable to highlight the complications in analysing scenarios with land-use change and non-$CO_2$ forcings. If such biogeochemically coupled simulations of SSP scenarios were to be included in future C4MIP phases one might need to request additional model output or request additional simulations.

*l 825: "these metrics become difficult to interpret": discuss alternative approaches proposed in Chimuka et al., 2023.*

We have included a discussion of the Chimuka et al. approach in the revised version of our summary and conclusions.

References

Boer, G. J., and V. Arora, 2009: Temperature and concentration feedbacks in the carbon cycle. Geophys. Res. Lett., 36, https://doi.org/10.1029/2008GL036220.

Chimuka, V. R., Nzotungicimpaye, C.-M., and Zickfeld, K.: Quantifying land carbon cycle feedbacks under negative CO2 emissions, Biogeosciences, 20, 2283–2299, https://doi.org/10.5194/bg-20-2283-2023, 2023.

Schwinger, J., and J. Tjiputra, 2018: Ocean Carbon Cycle Feedbacks Under Negative Emissions. Geophys. Res. Lett., 45, 5062–5070, https://doi.org/10.1029/2018GL077790.